# STEPTOOL: A STEP-GRAINED REINFORCEMENT LEARNING FRAMEWORK FOR TOOL LEARNING IN LLMS

## ABSTRACT

Despite having powerful reasoning and inference capabilities, Large Language Models (LLMs) still need external tools to acquire real-time information or domain-specific expertise to solve complex tasks, which is referred to as tool learning. Existing tool learning methods primarily rely on tuning with expert trajectories, focusing on token-sequence learning from a linguistic perspective. However, there are several challenges: 1) imitating static trajectories limits their ability to generalize to new tasks. 2) even expert trajectories can be suboptimal, and better solution paths may exist. In this work, we introduce StepTool, a novel step-grained reinforcement learning framework to improve tool learning in LLMs. It consists of two components: Step-grained Reward Shaping, which assigns rewards at each tool interaction based on tool invocation success and its contribution to the task, and Step-grained Optimization, which uses policy gradient methods to optimize the model in a multi-step manner. Experimental results demonstrate that StepTool significantly outperforms existing methods in multi-step, tool-based tasks, providing a robust solution for complex task environments.

## 1 INTRODUCTION

Large Language Models (LLMs) have demonstrated remarkable abilities in reasoning and inference, leading to impressive performance across a wide range of tasks (Brown et al., 2020; Zeng et al., 2022; OpenAI, 2023). However, some complex tasks that require real-time information or domain-specific knowledge often exceed the capacities of LLMs alone. In recent years, tool learning (Qin et al., 2024; Patil et al., 2023; Qin et al., 2023) has emerged as a promising solution by augmenting LLMs with external tools (APIs). As shown in Figure 1, LLMs can dynamically select, invoke, and interact with tools to receive real-time responses. After multi-step interactions with external tools, LLMs can effectively gather the necessary information to complete complex and challenging tasks.

To enhance the tool-learning capabilities of LLMs, most approaches rely on Supervised Fine-Tuning (SFT) (Qin et al., 2023; Patil et al., 2023), in which LLMs are trained to imitate expert-generated trajectories in a text generation manner. Each trajectory is a sequence composed of a user's query, multiple tool-callings and responses, illustrated in Figure 1. Despite its straightforward implementation, SFT encounters two key limitations in training LLMs for tool learning. Firstly, imitating static pre-defined tool sequences limits the model's ability to adapt to new tasks or environments. Secondly, expert trajectories can successfully complete tasks but may not be the optimal sequence of tool invocations. Blindly imitating these trajectories can lead to suboptimal task-solving performance.

In addition to SFT, we propose using Reinforcement Learning (RL) as another strategy for tool learning, offering a more dynamic perspective by treating tool learning as a sequential decision-making process. Under the RL perspective, each step of tool invocation is considered as an action that leads to a state transition, and models are trained from the action-state transitions. Previous works have explored applying RL to optimize LLMs in aligning with human preferences (RLHF) (Christiano et al., 2017; Ouyang et al., 2022) or mathematical reasoning (Lightman et al., 2023; Wang et al., 2023; Shao et al., 2024). Nevertheless, these methods are not well-suited for tool learning due to several key challenges: 1) Tool learning involves multiple decision steps and real-time feedback from external tools and environments. In contrast, RLHF is single-step based, and the steps in mathematical

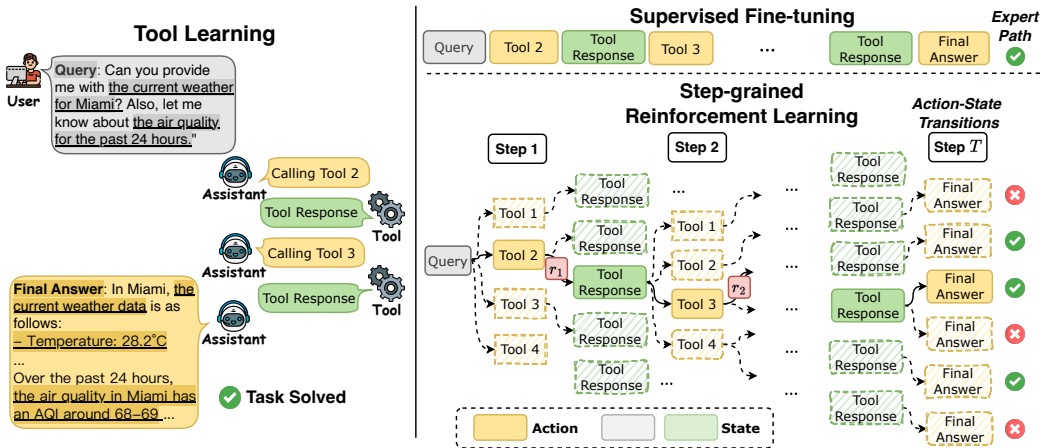

Figure 1: Tool learning scenario (left) and overall comparison between Supervised Fine-Tuning and Step-grained Optimization (right). SFT imitates expert trajectories, while the step-grained method utilizes step-level rewards to learn action-state transitions for optimization.

reasoning tasks are generated by the LLM itself, without feedback from the environment. 2) The reward of each step in tool learning is more complex, as it should consider not only the success of the tool invocation but also its contribution to task completion.

To harness the potential of RL in tool learning with multi-step environment interactions and address the limitations of existing methods, we propose **StepTool**, a novel step-grained reinforcement learning framework for tool learning, which models tool learning as a sequential decision-making process and treats each tool interaction as a critical decision point that directly impacts task completion, as shown in Figure 1. Specifically, StepTool consists of two core components: Step-grained Reward Shaping and Step-grained Optimization. For Step-grained Reward Shaping, we design rewards at each step based on both the accuracy of tool invocation and the contribution to the overall task completion, taking into account characteristics of intermediate actions in this scenario, i.e., well-defined formats and explicit task objectives. These step-grained rewards offer richer signals for tool learning, effectively guiding the model in decision-making. For Step-grained Optimization, we propose a step-grained reinforcement-based optimization method based on the theory of policy gradient (Williams, 1992; Sutton et al., 1999). This method ensures adaptability to dynamic, multi-step interactions, addressing the limitations of single-step approaches like RLHF.

In summary, this work makes the following contributions:

- We identify the limitations of static supervised fine-tuning (SFT) and the unsuitability of classic RLHF for tool learning, and introduce StepTool, a novel step-grained reinforcement learning framework. StepTool considers tool learning as a multi-step decision-making process, enabling models to learn from action-state transitions with real-time environment feedback.

- We design step-grained rewards tailored to tool learning scenarios, focusing on both the accuracy of tool invocation and the contribution to the overall task. These richer signals guide the model's decision-making. Additionally, we propose a step-grained optimization method based on policy gradients, ensuring adaptability to dynamic, multi-step interactions.

- Comprehensive experiments with three open-sourced models demonstrate the effectiveness of StepTool, confirming its superiority in enhancing the performance of solving complex tasks.

## 2 RELATED WORK

**Tool Learning.** Recent advancements in tool-augmented LLMs have expanded their ability to utilize external tools for complex tasks. Early research (Chen et al., 2023; Shen et al., 2024; Schick et al., 2024) propose to enable LLMs to interact with diverse external tools like program executors, search engines, and QA system. Building on these initial efforts, subsequent models have focused on more extensive interactions with real-world APIs and tools. Qin et al. (2023); Patil et al. (2023)

incorporate vast APIs from platforms like RapidAPI and TorchHub, training LLaMA model (Touvron et al., 2023) for tool-based tasks in a Supervised Fine-Tuning (SFT) manner. Additionally, some research efforts have concentrated on constructing verifiable and diverse datasets for SFT training Tang et al. (2023); Abdelaziz et al. (2024); Liu et al. (2024). Concurrent research (Chen et al., 2024) has explored the use of Direct Preference Optimization (DPO) (Rafailov et al., 2024) for Tool Learning. However, this approach constructs preference data pairs based on task completion, without accounting for the quality of intermediate steps. In contrast, our work explicitly shapes step-grained rewards and leverages them for step-grained reinforced optimization.

**Process Supervision in LLMs.** Process supervision has been extensively explored to enhance long-chain reasoning in LLMs. Previous studies (Lightman et al., 2023; Uesato et al., 2022; Ma et al., 2023; Shao et al., 2024; Wang et al., 2023) leverage pre-trained process reward models and optimize reasoning using RLHF (Ouyang et al., 2022). Recent advancements, such as step-level preferences in mathematical reasoning (Lai et al., 2024), apply DPO using step-level correctness. Unlike these works, our approach focuses on tool learning, where steps involve real-time interactions with external tools rather than text-based reasoning.

**Reinforcement Learning for Multi-Step Textual Tasks.** Recent advancements (Carta et al., 2023; Tan et al., 2024; Zhou et al., 2024; Wen et al., 2024) apply reinforcement learning (RL) to align LLMs for multi-step textual tasks. Carta et al. (2023); Tan et al. (2024) typically constrain the action space to a restricted subset, focusing on optimizing actions as a whole. In contrast, StepTool tackles tool learning, where the action space is expansive, involving complex and lengthy responses from environments. Examples of scenarios, illustrating the differences between task types, are provided in Appendix F. Methodologically, while prior works (Zhou et al., 2024; Wen et al., 2024) often rely on action-level models like Q-functions $Q(s, a)$ and value functions $V(s)$ to estimate intra-action influences, StepTool directly computes token-level advantages, capturing intra-action and inter-action influences without requiring action-level estimations. This approach avoids the inaccuracies of action-level models, offering a more efficient optimization framework for tool learning.

# 3 PROBLEM FORMULATION

In this work, we propose to model the tool learning process in LLMs as a multi-step decision-making problem, which can be formulated as a Markov Decision Process (MDP). The MDP is represented by the tuple $M = (\mathcal{S}, \mathcal{A}, \mathcal{P}, R, \gamma)$, with the following meanings:

- $\mathcal{S}$: The state space, where each state $s_t \in \mathcal{S}$ represents the current context or environment responses at time step $t$, in connection with prior tool interactions.

- $\mathcal{A}$: The action space, where each action $a_t \in \mathcal{A}$ corresponds to calling an external tool (API) or generating a final response (as a terminal action) at time $t$.

- $\mathcal{P}$: The state transition dynamics, $P(s_{t+1}|a_t, s_t)$ defines the probability of transitioning to a new state $s_{t+1}$ given the current state $s_t$ and the action $a_t$, representing how the environment changes as tools are applied.

- $\mathcal{R}$: The reward function, which assigns rewards $r_t = R(s_t, a_t)$ based on the current state $s_t$ and action $a_t$, representing the effectiveness of this tool-calling step.

- $\gamma$: The discount factor, which determines how the model balances immediate rewards with long-term task-solving performance.

Here we formulate the tool selection strategy of LLM as a decision-making policy $\pi_\theta$, parameterized by $\theta$, which governs the selection of actions (tools) based on the current state. A trajectory $\tau = \{s_1, a_1, s_2, a_2, ..., s_T, a_T\}$ represents a sequence of states and actions over time, reflecting the multiple interactions between LLMs and external tools or environments.

To maximize the final task-solving performance, the model seeks to optimize the expected reward $\overline{R_\theta}$, which is given by:

$$\overline{R_\theta} = \sum_\tau R(\tau)\pi_\theta(\tau) = \mathbb{E}_{\tau \sim \pi_\theta(\tau)} \left[ R(\tau) \right], \tag{1}$$

where $R(\tau)$ represents the reward for a given trajectory $\tau$, and $\pi_\theta(\tau)$ defines the probability of generating that trajectory under the policy $\pi_\theta$. The gradient of the expected reward can be computed to update the model's parameters (Williams, 1992), thereby enhancing the task-solving capabilities of the LLM:

$$
\begin{aligned}
\nabla \overline{R_\theta} = \sum_\tau R(\tau) \nabla \pi_\theta(\tau) &= \sum_\tau R(\tau) \pi_\theta(\tau) \nabla \log \pi_\theta(\tau) \\
&= \mathbb{E}_{\tau \sim \pi_\theta(\tau)} \left[ R(\tau) \nabla \log \pi_\theta(\tau) \right] \\
&= \mathbb{E}_{\tau \sim \pi_\theta(\tau), (s_t, a_t) \sim \tau} \left[ R(\tau) \sum_{t=1}^{T} \nabla \log \pi_\theta(a_t | s_t) \right].
\end{aligned}
\tag{2}
$$

To enhance learning efficiency and stabilize training, we replace $R(\tau)$ with the advantage function $\hat{A}(s_t, a_t)$ as most policy-gradient-based RL algorithms (Williams, 1992; Schulman et al., 2017) did, which measures the relative benefit of a given action compared to the expected return of the state:

$$
\hat{A}(s_t, a_t) = G_t^n - V(s_t) = r_t + \gamma r_{t+1} + \gamma^2 r_{t+2} + \ldots + \gamma^{T-t} r_T - V(s_t),
\tag{3}
$$

where $G_t^n$ represents the estimated future reward, and $V(s_t)$ is the value function, estimating the expected return when starting from state $s_t$ and following the current policy thereafter.

## 4 METHOD

Aimed at enhancing LLMs' ability to use multiple tools for complex task solving, we propose a novel step-grained reinforcement learning framework, **StepTool**, which is designed around the core principles of the advantage function (Equation 3) and the policy gradient formulation (Equation 2).

As illustrated in Figure 2, StepTool consists of two primary components: Step-grained Reward Shaping and Step-grained Optimization. Step-grained Reward Shaping assigns rewards at each tool interaction step, evaluating both the accuracy of tool invocation and the contribution to the overall task completion. Step-grained Optimization applies policy gradient methods to optimize the model in a multi-step manner. Together, these components provide step-grained feedback and optimize multi-step decisions, enhancing task-solving performance in complex environments.

### 4.1 STEP-GRAINED REWARD SHAPING

Step-grained Reward Shaping provides step-level reward signals for intermediate steps, effectively guiding the model in decision-making. In tool learning scenarios, the steps of tool invocation are characterized by well-defined formats and explicit task-oriented goals, naturally lending themselves to easier step-grained reward shaping. These step-grained rewards offer explicit feedback for each action, addressing the limitations of delayed rewards.

#### 4.1.1 STEP-GRAINED REWARD DESIGN

Considering well-defined formats and explicit task objectives of intermediate tool-calling actions, we have designed two key factors: the success of the tool call action (abbreviated as SuccCalling), and the contribution to the overall task completion (abbreviated as Contribution). For the final step, we directly link the reward to the completion of the task (abbreviated as IsSolved), representing to whether the user's query is solved.

**SuccCalling.** The SuccCalling metric evaluates whether the model successfully executes a tool call with the correct format and content ( i.e. tool name and arguments). SuccCalling can be formally represented as $\hat{r}_t^{\mathrm{SC}} = \mathrm{SuccCalling}(a_t, s_{t+1})$, where the reward at time $t$ is determined by the action $a_t$ and the subsequent state $s_{t+1}$.

However, simply making a correct tool call does not guarantee progress toward solving the task. To further guide the model, we introduce the Contribution metric, which evaluates how much the tool's action aids the overall task solution.

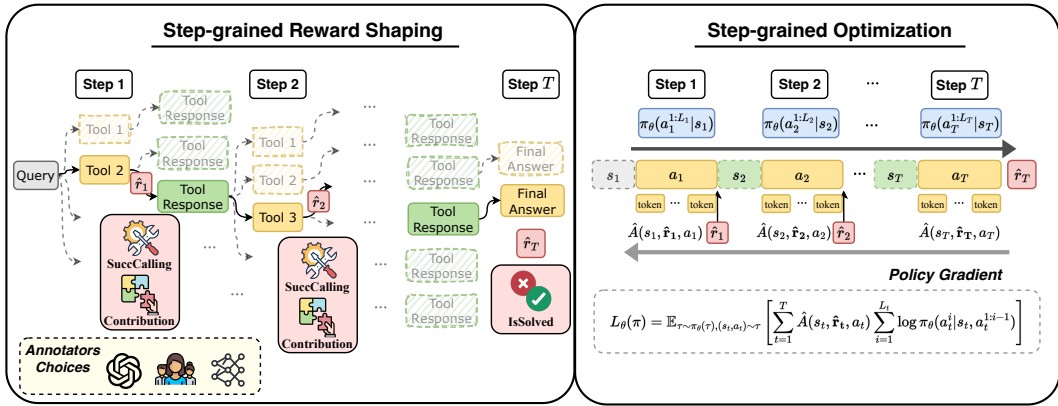

Figure 2: The architecture of **StepTool**, a step-grained reinforcement learning framework, featuring Step-grained Reward Shaping for assigning rewards at each tool interaction and Step-grained Optimization for refining decision-making based on policy gradient.

**Contribution.** The Contribution metric evaluates the extent to which the tool's action facilitates the overall task solution. Actions that contribute minimally, such as redundant steps or irrelevant outputs, receive lower rewards. The $Contribution$ score is based on the relationship between the current action and the final task-solving action, formally defined as $\hat{r}_t^{\text{Con}} = \text{Contribution}(a_t, a_T)$.

**IsSolved.** For the final step, the reward is directly associated with whether the task has been successfully completed. The IsSolved metric evaluates the final answer based on the initial user query, represented as $\hat{r}_t^{\text{IS}} = \text{IsSolved}(q, a_T)$. This reward only depends on the final step and the correctness of the response in addressing the user's query.

Formally, the reward for each action at step $t$ is defined as:

$$
\hat{r}_t = \begin{cases} \alpha \cdot \hat{r}_t^{\text{SC}} + \hat{r}_t^{\text{Con}} = \alpha \cdot \text{SuccCalling}(a_t, s_{t+1}) + \text{Contribution}(a_t, a_T), t = 1, 2, ..., T-1 \\ \hat{r}_t^{\text{IS}} = \text{IsSolved}(q, a_T), t = T, \end{cases}
$$

(4)

where $\alpha$ is a scaling factor to balance the weight of each component. To ensure consistency, rewards for both the intermediate steps and the final step are normalized to a uniform scale.

### 4.1.2 STEP-GRAINED REWARD ACQUISITION

To generate training data with step-grained rewards, we first collect multiple trajectories from the model's own inferences across tasks in the training set, each comprising multiple interactions between the model and external tools or environments. Step-grained rewards, derived from our reward components, can be assigned through various methods, such as automated rule-based models, human annotations, or advanced models like GPT-4 (OpenAI, 2023) (with the annotation prompts detailed in Appendix C). Considering the significant time and financial costs associated with human annotation, we primarily rely on a combination of rule-based systems and GPT-4 to handle the annotation process. These step-grained annotated data can be used for offline reinforcement learning optimization or to train a reward model for online training.

### 4.2 STEP-GRAINED OPTIMIZATION

Addressing the limitations of single-step approaches like RLHF (Ouyang et al., 2022), we propose a step-grained reinforced optimization strategy based on policy gradient that optimizes all prior steps, ensuring adaptability to dynamic, multi-step interactions.

### 4.2.1 STEP-GRAINED OPTIMIZATION OBJECTIVE

Building on the problem formulation (section 3), we now extend the gradient of the expected reward to a token-level consideration. Assumed each action $a_t$ consists of a sequence of $L_t$ tokens, the gradient of the expected return $\overline{R_\theta}$ at the step level is expressed as:

$$\nabla \overline{R_\theta} = \mathbb{E}_{\tau \sim \pi_\theta(\tau), (s_t, a_t) \sim \tau} \left[ \sum_{t=1}^{T} \hat{A}(s_t, a_t) \sum_{i=1}^{L_t} \nabla \log \pi_\theta(a_t^i | s_t, a_t^{1:i-1}) \right], \qquad (5)$$

where $\hat{A}(s_t, a_t)$ represents the advantage function for the action sequence $a_t$ at step $t$, which is composed of $L_t$ tokens. Through our step-grained reward shaping mechanism, we are able to calculate rewards at each time step $t$ in the trajectory. To better reflect the advantage of each action sequence, we implement the advantage function $\hat{A}(s_t, a_t)$ with our step-grained rewards $\hat{r}_t$ as:

$$\hat{A}(s_t, \hat{r}_t, a_t) = G_t^n - V(s_t) = \hat{r}_t + \gamma \hat{r}_{t+1} + \gamma^2 \hat{r}_{t+2} + \cdots + \gamma^{T-t} \hat{r}_T - V(s_t). \qquad (6)$$

The term $G_t^n$ reflects the cumulative future rewards based on these step-grained rewards $\hat{r}_t$, discounted by factor $\gamma$, extending from step $t$ onward, while $V(s_t)$ is the value function for the current state.

Our optimization objective is thus formalized as:

$$L_\theta(\pi) = \mathbb{E}_{\tau \sim \pi_\theta(\tau), (s_t, a_t) \sim \tau} \left[ \sum_{t=1}^{T} \hat{A}(s_t, \hat{r}_t, a_t) \sum_{i=1}^{L_t} \log \pi_\theta(a_t^i | s_t, a_t^{1:i-1}) \right]. \qquad (7)$$

This objective reflects the optimization of the policy $\pi_\theta$ by taking into account the step-level advantage with our step-grained rewards, encouraging the model to select actions that yield higher reward gains.

Additionally, it should be noted that classic RLHF (Ouyang et al., 2022) typically optimizes "prompt-response" data with final rewards based on human preferences, which is equivalent to treating the task as a single step ($T = 1$). However, in the scenario of tool learning involving multi-step interactions with external environments, each trajectory consists of multiple intermediate steps. Our method addresses the more complex case of $T > 1$ by applying step-grained rewards and optimizing actions at each step, ensuring both immediate and future outcomes are taken into account.

### 4.2.2 A PRACTICAL INSTANTIATION WITH PPO

Our framework is compatible with any policy gradient-based reinforcement learning algorithm. As a practical example, we implement the Proximal Policy Optimization (PPO) (Schulman et al., 2017) algorithm to demonstrate its application. Here, we estimate the advantage function using Generalized Advantage Estimation (GAE) to improve stability:

$$\begin{aligned} \hat{A}(s_t, \hat{r}_t, a_t) &= \delta_t + (\gamma\lambda)\delta_{t+1} + \cdots + (\gamma\lambda)^{T-t+1}\delta_{T-1}, \\ \delta_t &= \hat{r}_t + \gamma V(s_{t+1}) - V(s_t). \end{aligned} \qquad (8)$$

To achieve stable training, we employ the PPO-clip version, which introduces a clipping mechanism to prevent large updates during optimization. The loss function based on the clipped PPO objective is given by:

$$\begin{aligned} \mathcal{L}_\theta^{ppo}(\pi) = \hat{\mathbb{E}}_{\tau \sim \pi_\theta(\tau), (s_t, a_t) \sim \tau} \Bigg[ \min \Bigg( &\sum_{t=1}^{T} \hat{A}(s_t, \hat{r}_t, a_t) \sum_{i=1}^{L_t} \frac{\log \pi_\theta(a_t^i | s_t, a_t^{1:i-1})}{\log \pi_{\theta'}(a_t^i | s_t, a_t^{1:i-1})}, \\ &\sum_{t=1}^{T} \hat{A}(s_t, \hat{r}_t, a_t) \sum_{i=1}^{L_t} \text{clip} \Bigg( \frac{\log \pi_\theta(a_t^i | s_t, a_t^{1:i-1})}{\log \pi_{\theta'}(a_t^i | s_t, a_t^{1:i-1})}, 1 - \epsilon, 1 + \epsilon \Bigg) \Bigg) \Bigg], \end{aligned} \qquad (9)$$

where $\pi_{\theta'}$ represents the represents the old policy used to generate the previous trajectories, and $\epsilon$ is a hyperparameter that controls the allowable deviation between the current and old policies.

To further stabilize training, we also introduce a per-token KL divergence penalty from the old policy at each token, as proposed in RLHF (Ouyang et al., 2022). This helps to prevent large policy shifts during optimization. For our experiments, we apply the PPO version of our framework, which ensures robust performance in multi-step tool-based tasks.

## 5 EXPERIMENTS

### 5.1 EXPERIMENTAL SETTINGS

**Benchmark & Evaluation Metrics.**   We use StableToolBench (Guo et al., 2024), an improved version of ToolBench (Qin et al., 2023), consisting of 765 tasks across six subsets with varying tool categories and complexities. We applied two key metrics provided by this benchmark for evaluation: pass rate, measuring the proportion of tasks the model solves, and win rate, indicating how often our method outperforms baselines.

Table 1: Statistics of test tasks in StableToolBench. **Ins.**, **Cat.** and **Tool** stand for the Instruction, Category, and Tool subgroup in the test set, respectively.

|                   | I1 Ins. | I1 Cat. | I1 Tool | I2 Cat. | I2 Ins. | I3 Ins. |
|-------------------|---------|---------|---------|---------|---------|---------|
| **# Tasks**       | 163     | 153     | 158     | 106     | 124     | 61      |
| **# Candidate API** | 862   | 644     | 794     | 728     | 690     | 352     |
| **# Relevant API** | 371    | 328     | 358     | 301     | 261     | 180     |

**Baselines.**   Tool learning is an emerging area where most existing work relies on supervised fine-tuning (SFT) to enhance the tool-using capabilities of LLMs (Qin et al., 2023; Patil et al., 2023; Abdelaziz et al., 2024). While these works vary in dataset construction, we adopt **SFT** as a baseline using the same training data. As one of the first works introducing RL-based optimization for tool learning, relevant baselines are limited. We implemented a classic **RLHF-PPO** baseline, adapting RLHF (Ouyang et al., 2022) to tool learning tasks, designed to handle single-step data. We evaluated our framework on three open-source models: ToolLLaMA-2-7b-v2(ToolLlama) (Qin et al., 2023), Llama3.1-8B-Instruct (Llama3.1) (Touvron et al., 2023), and Qwen2-7B-Instruct (Qwen2) (Yang et al., 2024), using two strategies: Chain of Thought (CoT) (Wei et al., 2022) and Depth-First Search Decision Tree (DFSDT) (Qin et al., 2023). To ensure fairness in data origins, we excluded Direct Preference Optimization (DPO) Rafailov et al. (2024) due to the requirement for constructing comparative data.

**Training Setting.**   For SFT, Llama3.1 and Qwen2 are trained with static expert paths from GPT-4 (OpenAI, 2023), with training tasks sampled from ToolBench (Qin et al., 2023). ToolLlama is directly applied as it had already been pre-trained through similar manner. For RLHF-PPO and our Step-Tool, we obtain responses and interaction paths generated by each model towards user query samples of $5,000$ training tasks. We use both rule-based models and GPT-4 (gpt-4-turbo-2024-04-09) to annotate step-grained. More details of experiment settings can be found at Appendix B. For a fair comparison, we optimize all models with the default learning rate of $1e^{-5}$, batch size 8, and an initial KL coefficient 0.3 in the same experimental environment with four NVIDIA A100 GPUs.

### 5.2 MAIN RESULTS

Table 2 presents the performance comparison of StepTool with SFT and RLHF-PPO across three base models and two strategies, including the closed-source model gpt-3.5-turbo-0125 as a reference benchmark. Below are some key observations:

- StepTool consistently outperforms SFT and RLHF-PPO across most subsets for the same base model and strategy, demonstrating the effectiveness of StepTool. Notably, under the DFSDT strategy on Qwen2, StepTool achieves a pass rate of over 60% on all subsets except for 'I2 Ins.'.

- The improvement varies across subsets. For simpler subsets like I1 Tool,' StepTool shows moderate gains of 1%-4%, whereas for more complex subsets like 'I3 Ins.,' improvements range from 5%-13%. It indicates StepTool's strength in handling tasks involving multiple tools and categories.

- StepTool generates better solution paths, as indicated by the win rate metric. Figure 3 shows StepTool's win rates against baselines across three subsets, consistently outperforming SFT and RLHF-PPO on ToolLLaMA with win rates from 50% to 65.8%, further demonstrating its advantage in tool-based task solving.

To further understand the model's behavior, we also conducted experiments on step length to evaluate its impact on planning and task decomposition. More detailed analysis can be found in Appendix A.

Table 2: Performance comparison between StepTool and other baselines on Pass Rate. We run all models once and take the average results from three times evaluations. StepTool performs best most of the time.

| BaseModel | Strategy | Method | Pass Rate (%) | | | | | | |
|---|---|---|---|---|---|---|---|---|---|
| | | | I1 Ins. | I1 Cat. | I1 Tool | I2 Cat. | I2 Ins. | I3 Ins. | Average |
| GPT-3.5 | CoT | / | $53.8_{\pm1.2}$ | $48.0_{\pm0.7}$ | $51.4_{\pm1.2}$ | $55.5_{\pm1.2}$ | $43.4_{\pm1.3}$ | $53.8_{\pm0.4}$ | $51.0_{\pm1.0}$ |
| | DFSDT | / | $60.0_{\pm0.5}$ | $53.5_{\pm1.3}$ | $65.7_{\pm0.5}$ | $61.6_{\pm1.2}$ | $50.5_{\pm0.7}$ | $65.6_{\pm2.7}$ | $59.5_{\pm1.2}$ |
| ToolLlama | CoT | / | $54.2_{\pm0.5}$ | $50.3_{\pm0.8}$ | $56.5_{\pm1.5}$ | $52.0_{\pm0.6}$ | $45.4_{\pm0.6}$ | $37.2_{\pm1.0}$ | $49.3_{\pm0.8}$ |
| | | RLHF-PPO | $55.0_{\pm1.9}$ | $50.5_{\pm0.9}$ | $42.3_{\pm0.7}$ | $46.4_{\pm0.7}$ | $42.1_{\pm1.6}$ | $35.2_{\pm1.2}$ | $45.3_{\pm1.2}$ |
| | | StepTool | $\mathbf{58.7}_{\pm1.8}$ | $\mathbf{57.8}_{\pm1.7}$ | $57.2_{\pm0.7}$ | $52.7_{\pm0.8}$ | $52.7_{\pm1.0}$ | $42.1_{\pm1.5}$ | $53.5_{\pm1.3}$ |
| | DFSDT | / | $57.0_{\pm1.0}$ | $52.3_{\pm1.5}$ | $57.5_{\pm1.2}$ | $52.4_{\pm0.7}$ | $49.7_{\pm1.7}$ | $53.8_{\pm1.9}$ | $53.8_{\pm1.3}$ |
| | | RLHF-PPO | $57.5_{\pm1.5}$ | $54.2_{\pm0.5}$ | $53.5_{\pm2.0}$ | $50.8_{\pm1.2}$ | $48.1_{\pm0.8}$ | $43.2_{\pm0.4}$ | $51.2_{\pm1.1}$ |
| | | StepTool | $\mathbf{59.7}_{\pm0.5}$ | $\mathbf{55.9}_{\pm0.0}$ | $\mathbf{58.4}_{\pm1.2}$ | $\mathbf{52.8}_{\pm1.2}$ | $\mathbf{51.3}_{\pm0.2}$ | $\mathbf{66.7}_{\pm0.4}$ | $\mathbf{57.5}_{\pm0.6}$ |
| Llama3.1 | CoT | SFT | $53.9_{\pm1.2}$ | $52.6_{\pm1.4}$ | $51.9_{\pm0.9}$ | $52.2_{\pm1.7}$ | $44.7_{\pm0.4}$ | $36.3_{\pm0.8}$ | $48.6_{\pm1.1}$ |
| | | RLHF-PPO | $50.2_{\pm0.9}$ | $\mathbf{57.8}_{\pm0.8}$ | $53.0_{\pm0.6}$ | $52.3_{\pm1.6}$ | $49.2_{\pm1.5}$ | $38.0_{\pm1.5}$ | $50.1_{\pm1.2}$ |
| | | StepTool | $\mathbf{54.3}_{\pm1.0}$ | $56.4_{\pm0.3}$ | $\mathbf{53.2}_{\pm0.9}$ | $\mathbf{53.9}_{\pm1.7}$ | $\mathbf{49.7}_{\pm0.8}$ | $\mathbf{42.6}_{\pm2.4}$ | $\mathbf{51.7}_{\pm1.2}$ |
| | DFSDT | SFT | $58.8_{\pm1.2}$ | $58.0_{\pm1.6}$ | $59.8_{\pm0.9}$ | $53.9_{\pm1.9}$ | $53.5_{\pm0.9}$ | $45.9_{\pm1.3}$ | $55.0_{\pm1.3}$ |
| | | RLHF-PPO | $58.9_{\pm0.7}$ | $\mathbf{61.4}_{\pm0.7}$ | $59.9_{\pm1.0}$ | $55.9_{\pm1.0}$ | $49.5_{\pm0.0}$ | $44.8_{\pm0.4}$ | $55.1_{\pm0.9}$ |
| | | StepTool | $\mathbf{59.3}_{\pm0.8}$ | $60.9_{\pm1.3}$ | $\mathbf{60.2}_{\pm1.3}$ | $\mathbf{56.2}_{\pm1.6}$ | $\mathbf{59.3}_{\pm1.4}$ | $\mathbf{50.5}_{\pm1.0}$ | $\mathbf{57.7}_{\pm1.2}$ |
| Qwen2 | CoT | SFT | $53.0_{\pm0.6}$ | $54.5_{\pm0.7}$ | $59.9_{\pm1.2}$ | $54.0_{\pm0.3}$ | $\mathbf{45.6}_{\pm1.4}$ | $40.7_{\pm0.8}$ | $51.3_{\pm0.8}$ |
| | | RLHF-PPO | $58.8_{\pm0.9}$ | $54.9_{\pm0.7}$ | $57.0_{\pm0.5}$ | $54.3_{\pm1.0}$ | $45.1_{\pm1.0}$ | $48.4_{\pm3.1}$ | $53.1_{\pm1.2}$ |
| | | StepTool | $\mathbf{59.6}_{\pm1.1}$ | $\mathbf{56.1}_{\pm0.8}$ | $\mathbf{61.8}_{\pm0.8}$ | $\mathbf{54.8}_{\pm0.6}$ | $44.5_{\pm2.6}$ | $\mathbf{48.6}_{\pm1.9}$ | $\mathbf{54.2}_{\pm1.3}$ |
| | DFSDT | SFT | $63.7_{\pm1.3}$ | $59.3_{\pm1.3}$ | $64.8_{\pm1.0}$ | $56.7_{\pm1.1}$ | $49.1_{\pm2.1}$ | $57.7_{\pm1.0}$ | $58.6_{\pm1.3}$ |
| | | RLHF-PPO | $64.1_{\pm0.3}$ | $58.9_{\pm2.4}$ | $66.9_{\pm2.2}$ | $59.8_{\pm0.8}$ | $49.8_{\pm1.2}$ | $54.4_{\pm1.7}$ | $59.0_{\pm1.4}$ |
| | | StepTool | $\mathbf{65.6}_{\pm1.8}$ | $\mathbf{60.8}_{\pm0.3}$ | $\mathbf{68.4}_{\pm1.6}$ | $\mathbf{60.9}_{\pm0.9}$ | $\mathbf{51.1}_{\pm1.8}$ | $\mathbf{65.3}_{\pm1.7}$ | $\mathbf{62.0}_{\pm1.4}$ |

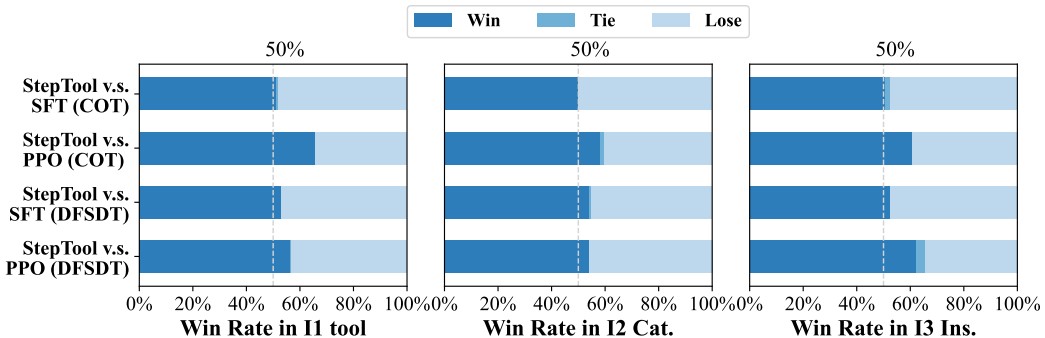

Figure 3: Win rates of StepTool against other methods based on ToolLlama across three randomly selected subsets. StepTool has a win rate over 50% against all baselines.

## 5.3 PASS@K: ASSESSING KNOWLEDGE DISCOVERY VS. PRIOR RE-WEIGHTING

We computed Pass@k metrics, widely used in domains like mathematical reasoning (Ni et al., 2022; Havrilla et al., 2024), to assess whether StepTool enables models to discover new knowledge or merely re-weight prior knowledge. Experiments were conducted on ToolLlama before and after StepTool optimization under CoT strategy, sampling 8 trajectories per task under a temperature setting of 0.7. Due to the time cost of real-world API interactions, 20 tasks from each StableToolBench subsets (Guo et al., 2024) were randomly selected, with results averaged over three independent evaluations.

As shown in Table 3, ToolLlama optimized with StepTool outperforms ToolLlama across Pass@2, Pass@4, and Pass@8 metrics in most experimental settings. The improved Pass@k scores (across all values of k) suggest that the model is not merely re-weighting its prior knowledge but is also benefiting from the discovery of new knowledge during RL optimization.

Table 3: Pass@k performance comparison between ToolLlama with and without StepTool. Experiments were conducted under CoT strategy, sampling 8 trajectories per task.

| BaseModel | Method | I1 Ins. | I1 Cat. | I1 Tool | I2 Cat. | I2 Ins. | I3 Ins. | Average |
|---|---|---|---|---|---|---|---|---|
| | | | | *Pass@2* | | | | |
| ToolLlama | / | $58.3_{\pm3.1}$ | $\mathbf{54.2}_{\pm5.4}$ | $51.7_{\pm2.4}$ | $50.0_{\pm0.0}$ | $50.8_{\pm5.1}$ | $55.0_{\pm1.0}$ | $53.3_{\pm2.8}$ |
| | + StepTool | $\mathbf{58.3}_{\pm1.2}$ | $53.3_{\pm2.4}$ | $\mathbf{70.8}_{\pm4.2}$ | $\mathbf{53.3}_{\pm2.4}$ | $\mathbf{68.3}_{\pm2.4}$ | $\mathbf{60.0}_{\pm4.1}$ | $\mathbf{60.7}_{\pm2.8}$ |
| | | | | *Pass@4* | | | | |
| ToolLlama | / | $\mathbf{65.8}_{\pm4.2}$ | $60.0_{\pm5.4}$ | $56.7_{\pm2.4}$ | $66.7_{\pm6.2}$ | $61.7_{\pm2.4}$ | $62.5_{\pm2.0}$ | $62.2_{\pm3.8}$ |
| | + StepTool | $65.0_{\pm2.0}$ | $\mathbf{61.7}_{\pm3.1}$ | $\mathbf{80.8}_{\pm4.2}$ | $\mathbf{67.5}_{\pm2.0}$ | $\mathbf{74.2}_{\pm1.2}$ | $\mathbf{70.0}_{\pm4.1}$ | $\mathbf{69.9}_{\pm2.8}$ |
| | | | | *Pass@8* | | | | |
| ToolLlama | / | $\mathbf{70.8}_{\pm4.2}$ | $65.0_{\pm5.4}$ | $61.7_{\pm2.4}$ | $\mathbf{68.3}_{\pm4.7}$ | $71.7_{\pm2.4}$ | $68.3_{\pm2.4}$ | $67.6_{\pm3.6}$ |
| | + StepTool | $66.7_{\pm1.2}$ | $\mathbf{70.0}_{\pm2.0}$ | $\mathbf{80.8}_{\pm4.2}$ | $67.5_{\pm3.1}$ | $\mathbf{79.2}_{\pm4.2}$ | $\mathbf{76.7}_{\pm5.1}$ | $\mathbf{73.5}_{\pm3.3}$ |

## 5.4 Ablation Study: Impact of Step-grained Components

To evaluate the contributions of each step-grained component in StepTool, we tested two variants: - *w/o Step-grained Reward*, where intermediate rewards are set to 0, and - *w/o Step-grained Opt*, where sub-trajectories ending with intermediate actions are optimized with PPO. As shown in Table 4, removing either step-grained rewards or step-grained optimization results in significant performance degradation. These results highlight the importance of intermediate rewards for providing informative signals and the limitations of traditional RLHF-PPO in capturing step dependencies. Both components are critical for the effectiveness of our framework in solving multi-step tasks.

Table 4: Ablation study on two components of StepTool. Eliminating each component leads to reduced performance.

| Method | Pass Rate (%) | | | | | | |
|---|---|---|---|---|---|---|---|
| | I1 Ins. | I1 Cat. | I1 Tool | I2 Cat. | I2 Ins. | I3 Ins. | Average |
| ToolLlama + StepTool | $\mathbf{58.7}_{\pm1.8}$ | $\mathbf{57.8}_{\pm1.7}$ | $\mathbf{57.2}_{\pm0.7}$ | $\mathbf{52.7}_{\pm0.8}$ | $\mathbf{52.7}_{\pm1.0}$ | $\mathbf{42.1}_{\pm1.5}$ | $\mathbf{53.5}_{\pm1.3}$ |
| - w/o Step-grained Reward | $57.2_{\pm2.6}$ | $50.5_{\pm0.4}$ | $45.1_{\pm0.8}$ | $44.9_{\pm1.5}$ | $51.1_{\pm1.5}$ | $39.9_{\pm0.8}$ | $48.1_{\pm1.3}$ |
| - w/o Step-grained Opt | $57.7_{\pm1.5}$ | $52.2_{\pm1.3}$ | $43.0_{\pm1.4}$ | $45.3_{\pm0.8}$ | $41.8_{\pm1.1}$ | $41.5_{\pm1.5}$ | $46.9_{\pm1.3}$ |
| ToolLlama | $54.2_{\pm0.5}$ | $50.3_{\pm0.8}$ | $56.5_{\pm1.5}$ | $52.0_{\pm0.6}$ | $45.4_{\pm0.6}$ | $37.2_{\pm1.0}$ | $49.3_{\pm0.8}$ |

## 5.5 Analysis of Tool Invocation Success Rates

To verify the effectiveness of our method in improving tool invocation during intermediate steps, we calculate the average success rates of tool invocations across all intermediate steps in the test sets for both ToolLLaMA and Qwen2 models. As illustrated in Figure 4, StepTool consistently improves the success rates of intermediate tool invocations in both CoT and DFSDT settings, demonstrating enhanced tool accuracy and effectiveness in multi-step tasks.

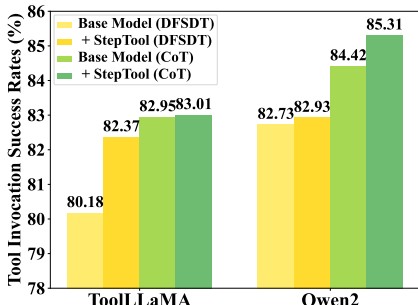

Figure 4: Tool invocation success rates for different methods using ToolLlama and Qwen2 base models.

## 5.6 Qualitative Analysis

We conducted a qualitative analysis to understand how StepTool improves intermediate actions. Figure 5 illustrates a case where StepTool corrects a wrong tool selection by ToolLlama. More examples are available in Appendix E. In this example, the user requests channel information, video comments, and streaming sources for movies. ToolLlama initially retrieves the correct channel info and video comments but mistakenly calls the 'getvideoscomment' tool again instead of switching to the 'download_stream' tool. After applying StepTool, the model correctly uses the 'download_stream' tool, providing the streaming link and fulfilling the request. This demonstrates StepTool's effectiveness in optimizing intermediate decisions for complex tasks.

## 6 CONCLUSION

In this work, we proposed StepTool, a novel step-grained reinforcement learning framework that enhances LLMs' ability to solve complex, multi-step tasks using multiple tools. StepTool features two core components: Step-grained Reward Shaping, which evaluates tool invocation success and task contributions, and Step-grained Optimization, which uses policy gradient methods to optimize decision-making at each step. Experiments with three open-source models demonstrate its effectiveness in improving task-solving performance. StepTool offers a fresh perspective on optimizing tool-use tasks, complementing SFT-based approaches and paving the way for future RL innovations tailored to tool learning.

## LIMITATIONS

Despite the superior performance of this work, several limitations must be acknowledged. First, consistent with prior studies (Shen et al., 2023; Rafailov et al., 2024), the PPO training process can be unstable. We have included all experimental setups and parameter settings in our code repository to ensure reproducibility. Second, despite achieving notable results, there is room for further improvement. Our method supports online multi-round data collection and optimization, but due to time and cost constraints, we conducted only a single round of offline training. This likely limited the model's full potential for performance gains.

---

**A Case of Correcting Wrong Tool Selection**

| ToolLlama | ToolLlama with StepTool |
|---|---|
| **User Query:** I'm planning a movie night with my friends and I need some recommendations. Get me the channel info for 'Paramount Pictures', and the comments for the video with ID '123456'. Also, find a website where I can stream or download the movie with the ID 'UxxajLWwzqY'. Begin! | **User Query:** I'm planning a movie night with my friends and I need some recommendations. Get me the channel info for 'Paramount Pictures', and the comments for the video with ID '123456'. Also, find a website where I can stream or download the movie with the ID 'UxxajLWwzqY'. Begin! |
| **Step 1:** Tool: getchannelinfo
Args: {"channel_id": "Paramount Pictures"}
**Tool Response:** Paramount Pictures is a major American film studio, offering ... | **Step 1:** Tool: getchannelinfo
Args: {"channel_id": "Paramount Pictures"}
**Tool Response:** Paramount Pictures is a major American film studio, offering ... |
| **Step 2:** Tool: getvideoscomment
Args: {"video_id": "123456"}
**Tool Response:** {"user": "alicewonderland", "comment": "Insightful video, learned a lot!", "timestamp": "2023-10-05T08:21:45Z" ... | **Step 2:** Tool: getvideoscomment
Args: {"video_id": "123456"}
**Tool Response:** {"user": "alicewonderland", "comment": "Insightful video, learned a lot!", "timestamp": "2023-10-05T08:21:45Z" ... |
| **Step 3:** Tool: getvideoscomment
Args: {"video_id": "UxxajLWwzqY"}
**Tool Response:** {"user": "video_fanatic", "comment": "This video blew my mind! The quality is top-notch.", "timestamp": "2023 ...
# Wrong Tool Selection | **Step 3:** Tool: download_stream
Args: {"is_id": "UxxajLWwzqY"}
**Tool Response:** {'id': 'UxxajLWwzqY', 'title': 'Icona Pop - I Love It (feat. Charli XCX) [OFFICIAL VIDEO]', DOWNLOAD ... https://smarturl.it/UxxajLWwzqY ...} |
| **Step 4:** Finish
...The channel info for 'Paramount Pictures'... The comments for the video with ID '123456' are as follows: 'Insightful video, learned a lot!' - alicewonderland, ... The comments for the video with ID 'UxxajLWwzqY' are as follows: 'This video blew my mind!... | **Step 4:** Finish
...The channel info for 'Paramount Pictures'...The comments for the vide with ID '123456' are as follows: 'Insightful video, learned a lot!'...Regarding the movie with ID 'UxxajLWwzqY', you can download it from https://smarturl.it/UxxajLWwzqY. Enjoy your movie night! |
| **Task Not Solved.** | **Task Solved.** |

Figure 5: A case study on the I3 Ins. subset comparing ToolLlama and ToolLlama with StepTool: StepTool successfully corrects the wrong tool selection.

REPRODUCIBILITY STATEMENT

To ensure reproducibility, we provide an anonymous GitHub repository containing all necessary implementation code for our method, as well as the experimental setups, model configurations, and scripts needed to reproduce our results. This repository can be accessed here: https://anonymous.4open.science/r/StepTool.

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

# A    ANALYSIS ON AVERAGE STEP LENGTH

Step length, or the number of tools utilized per task, can serve as a potential metric to evaluate both task difficulty and a model's planning capabilities. However, in the StableToolBench benchmark (Guo et al., 2024), most tasks are designed to require 2-3 API calls for successful completion, with 68.6% of tasks requiring 2 APIs and 19.9% requiring 3 APIs, accounting for 88.5% of all tasks. Task difficulty in this benchmark is primarily influenced by factors such as tool diversity (e.g., intra-tool, intra-category, or intra-collection variations) and whether the required tools are included in the training data (Qin et al., 2023). These design constraints ensure consistency in task complexity but may limit the ability of step length to fully capture a model's planning abilities.

Despite this, we measured the models' performance on the average number of tool utilized per task across different subsets. The results are summarized in Table 5.

Table 5: Comparison of average step length (number of tools utilized per task) across different subsets and methods.

| BaseModel | Strategy | Method | Step Length | | | | | | |
|---|---|---|---|---|---|---|---|---|---|
| | | | I1 Ins. | I1 Cat. | I1 Tool | I2 Cat. | I2 Ins. | I3 Ins. | Average |
| ToolLlama | CoT | SFT | 2.2025 | 2.1634 | 2.3608 | 2.4623 | 2.5565 | 2.5738 | 2.3866 |
| | | RLHF-PPO | 2.2025 | 2.0980 | 2.2911 | 2.4057 | 2.4919 | 2.5410 | 2.3384 |
| | | StepTool | 2.2515 | 2.1438 | 2.3228 | 2.4906 | 2.5081 | 2.4918 | 2.3681 |
| | DFSDT | SFT | 2.1595 | 2.0980 | 2.2025 | 2.3868 | 2.5161 | 2.5410 | 2.3173 |
| | | RLHF-PPO | 2.1779 | 2.0719 | 2.2342 | 2.3208 | 2.4516 | 2.4918 | 2.2914 |
| | | StepTool | 2.2025 | 2.1046 | 2.2595 | 2.4528 | 2.5081 | 2.3934 | 2.3202 |
| Llama3.1 | CoT | SFT | 2.1779 | 2.1438 | 2.1709 | 2.3962 | 2.4194 | 2.4918 | 2.3000 |
| | | RLHF-PPO | 2.1534 | 2.1176 | 2.2152 | 2.4151 | 2.3952 | 2.5574 | 2.3090 |
| | | StepTool | 2.1656 | 2.0458 | 2.1962 | 2.4245 | 2.3226 | 2.3770 | 2.2553 |
| | DFSDT | SFT | 2.1718 | 2.1438 | 2.1519 | 2.4151 | 2.3871 | 2.4262 | 2.2826 |
| | | RLHF-PPO | 2.1656 | 2.1176 | 2.2152 | 2.4245 | 2.3790 | 2.4754 | 2.2962 |
| | | StepTool | 2.1902 | 2.0131 | 2.1203 | 2.4245 | 2.2984 | 2.3934 | 2.2400 |
| Qwen2 | CoT | SFT | 2.2209 | 2.0850 | 2.2152 | 2.4340 | 2.4274 | 2.5738 | 2.3260 |
| | | RLHF-PPO | 2.2147 | 2.1699 | 2.2519 | 2.4434 | 2.4274 | 2.5574 | 2.3441 |
| | | StepTool | 2.1902 | 2.1634 | 2.2215 | 2.4717 | 2.4194 | 2.5902 | 2.3427 |
| | DFSDT | SFT | 2.1963 | 2.1242 | 2.2405 | 2.4057 | 2.4355 | 2.5082 | 2.3184 |
| | | RLHF-PPO | 2.2025 | 2.1503 | 2.2532 | 2.4245 | 2.4355 | 2.4754 | 2.3236 |
| | | StepTool | 2.1840 | 2.0980 | 2.1899 | 2.4057 | 2.4113 | 2.5902 | 2.3132 |

From the results, we observe the following key points:

- Minimal Variation: The average number of tool invocations per task shows minimal variation (<0.1 tool calls per task) across models under the same experimental settings. This observation is highly related to the task design in the benchmark, as mentioned above.
- Lack of Correlation with Performance: The number of tool invocations does not directly correlate with model performance. Fewer invocations may indicate missed critical API calls, while more invocations could reflect redundant or inefficient steps.

While step length offers insights into a model's planning and task decomposition capabilities, its utility is constrained by the current benchmark design, which emphasizes tool invocation accuracy over broader planning strategies. Future work could explore benchmarks with more diverse and open-ended task designs to better evaluate these aspects.

# B    IMPLEMENTATION DETAILS OF BASELINES

For fair comparison, we implemented RLHF-PPO as a baseline following the traditional RLHF framework (Ouyang et al., 2022). The PPO baseline is designed for single-step reward optimization, applying rewards only to the final step of a trajectory. In contrast, StepTool introduces step-grained reward integration to capture dependencies across the entire trajectory.

Both approaches were trained using the same dataset, reward annotations, and adaptive KL penalty settings (Ziegler et al., 2019). The initial KL coefficient was set to 0.3, with adjustments based on the adaptive KL controller. Additional hyperparameters, such as learning rate, batch size, and policy update frequency, were kept consistent across all methods to ensure a fair comparison.

The configuration file in our code repository provides further details on the experimental settings, including hyperparameter values, model initialization, and training schedules. These details are shared to enhance transparency and reproducibility.

## C    REFERENCE PROMPT FOR STEP-GRAINED REWARD ANNOTATION

Here we provide a reference prompt for GPT-4 to perform step-grained reward annotation:

---

**Instruction Prompt for Step-wise Reward**

Query:
{query}

Intermediate Steps:
{mid_steps}
Final Answer:
{final_answer}

Given the above query, all intermediate steps and the final answer, you need to evaluate the entire task-solving process by following rules:
(1) **Successful Tool Calling:** For each intermediate step, determine if a tool was called successfully and give a score of 0 (no) or 1 (yes).
(2) **Contribution to Final Answer:** For each intermediate step, rate its contribution to the final answer on a scale from 0 to 5.
(3) **Final Answer Status:** Determine if the final answer is "Solved", "Unsure", or "Unsolved".

Now provide your evaluation in JSON format with the parameters of "succeed_tool_calling", "contribution_to_final_answer" and "final_answer_status" to the function 'evaluate_process_reward".

---

Figure 6: A Reference Prompt for Step-grained Reward Annotation.

## D    DISCUSSION ON THE RISK OF REWARD HACKING

Reward hacking, which refers to potential annotation errors in this work, may arise during automated or manual labeling of auxiliary rewards or through unintended exploitation of reward structures by models. To enhance the robustness of reward annotations and mitigate these risks, several strategies can be considered:

- Reward Model Ensembles: Using multiple reward models and averaging their outputs can reduce the impact of individual model biases.
- Multiple Annotations: Aggregating annotations from diverse sources can provide a more balanced reward signal.
- Iterative Refinement: Refining reward annotations iteratively based on experimental results can improve alignment with the task's ultimate objectives.

These strategies, when implemented, can help create more robust reward systems, improving the reliability and robustness of reinforcement learning frameworks in complex tasks.

# E    MORE CASES

## E.1    A CASE OF CORRECTING MISSING RELEVANT TOOL INVOCATION

This case from I2 Ins. illustrates that StepTool improves tool invocation by correcting missing relevant tool invocation.

The user requests cocktail recipes with vodka and a cake recipe from the Birthday Cake DB. While both systems successfully retrieve the cocktail recipe, ToolLLaMA fails to invoke the tool for the detailed cake recipe, prematurely finishing the task. In contrast, ToolLLaMA with StepTool correctly calls the tool, retrieving the complete cake recipe and solving the task. This demonstrates StepTool's effectiveness in enhancing tool invocation accuracy for multi-step tasks.

---

**A Case of Correcting Missing Relevant Tool Invocation**

| **ToolLlama** | **ToolLlama with StepTool** |
|---|---|
| **User Query:** I'm hosting a cocktail party and I need some cocktail recipes. Can you suggest some cocktail recipes that contain vodka as one of the ingredients? Also, provide me with a cake recipe from The Birthday Cake DB.Begin! | **User Query:** I'm hosting a cocktail party and I need some cocktail recipes. Can you suggest some cocktail recipes that contain vodka as one of the ingredients? Also, provide me with a cake recipe from The Birthday Cake DB.Begin! |
| **Step 1:** Tool: search_by_ingredient_for_ cocktails Args: {"ingredient": "vodka"} **Tool Response:** ... | **Step 1:** Tool: search_by_ingredient_for_ cocktails Args: {"ingredient": "vodka"} **Tool Response:** ... |
| **Step 2:** Tool: list_of_cakes_for_the_birthday _cake_db Args: {} **Tool Response:** {"id": "1", "title": "Raspberry and custard muffins", "difficulty": "Easy", "image": "https://apipics.s3..."}, {"id": "2","title":"Lemon and blackberry...} ... | **Step 2:** Tool: list_of_cakes_for_the_birthday _cake_db Args: {} **Tool Response:** {"id": "1", "title": "Raspberry and custard muffins", "difficulty": "Easy", "image": "https://apipics.s3..."}, {"id":"2","title":"Lemon and blackberry...} ... |
| *# Miss A Relevant Tool Calling* | **Step 3:** Tool: detailed_cake_recipe_by_id Args: {"is_id": "1"} **Tool Response:** {'ingredients': ["140g unsalted butter, melted", "3 medium free-range eggs", "125g custard", ...], 'method': [{"Step 1": "Heat the oven to 170...}...]] |
| **Step 4:** Finish Sure! Here are some cocktail recipes that contain vodka as one of the ingredients: 1. Vodka Martini... lot!'...I can provide you with a cake recipe from The Birthday Cake DB. How about trying the Raspberry and custard muffins? It's an easy recipe and you can find the image here: https://apipics.s3... | **Step 4:** Finish Sure! Here are some cocktail recipes that contain vodka as one of the ingredients: 1. Vodka Martini... lot!'... And here is a cake recipe from The Birthday Cake DB: Raspberry and custard muffins ... Ingredients: -140g unsalted butter ... |
| **Task Not Solved.** | **Task Solved.** |

Figure 7: A case study on the I2 Ins. subset comparing ToolLlama and ToolLlama with StepTool: ToolLlama missed a relevant tool invocation while StepTool successfully called the relevant tool and completed the task.

## E.2    A CASE OF CORRECTING INCORRECT ARGUMENTS

This case from I2 Ins. illustrates that StepTool improves tool invocation by correcting incorrect Arguments.

In this case, the user requests books by a specific author (J.K. Rowling) and interesting space news. ToolLLaMA fails to retrieve the correct book information due to incorrectly formatted arguments passed to the tool, resulting in incomplete search results. However, ToolLLaMA with StepTool

successfully corrects the tool arguments, retrieving the correct list of books along with the requested space news, demonstrating StepTool's ability to effectively handle incorrect arguments and complete the task.

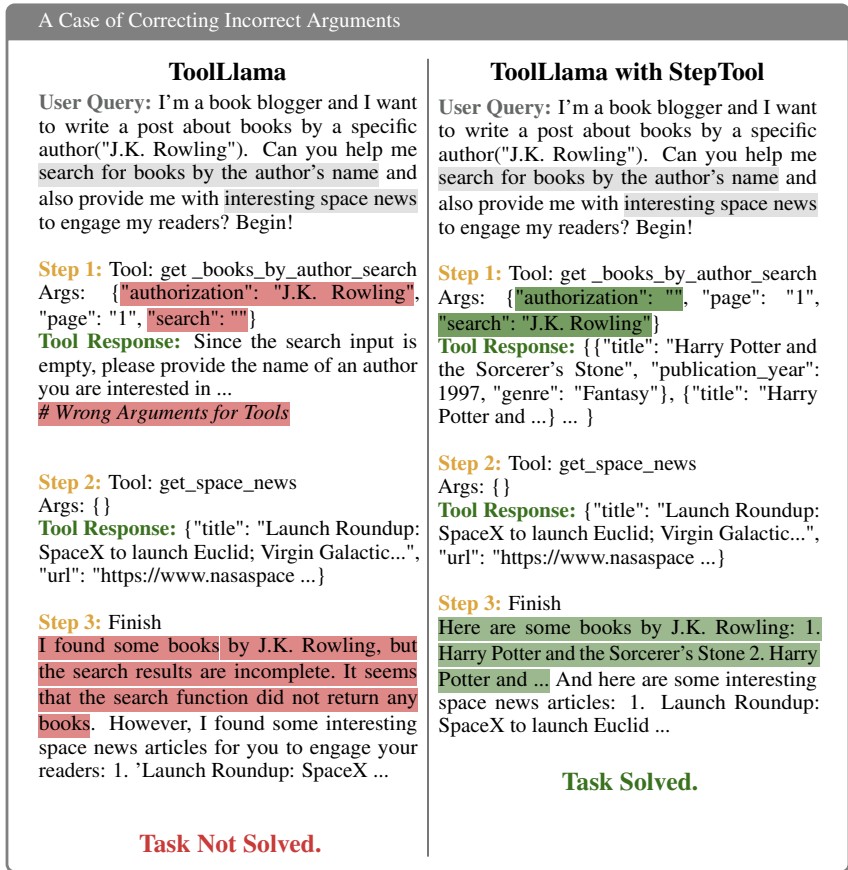

Figure 8: A case study on the I2 Ins. subset comparing ToolLlama and ToolLlama with StepTool: StepTool successfully corrects incorrect arguments during tool invocation.

## F    EXAMPLES OF TWENTY QUESTIONS AND TOOLBENCH

In Figure 9 and Figure 10, we compare two different task scenarios: Twenty Questions and ToolBench. While Twenty Questions involves a structured, short-action environment with clear, immediate rewards based on each question's effectiveness, ToolBench tasks, such as querying for cocktail recipes, require tool interactions with complex, lengthy outputs. In ToolBench, rewards are not immediate and must be carefully designed to reflect both intermediate tool usage and the final output's correctness, highlighting the added complexity of tool-based tasks.

**Assistant**: Is it an animal?
**Environment**: No.
**Assistant**: Is it an inanimate object?
**Environment**: Yes.
**Assistant**: Is it man-made?
**Environment**: No.
**Assistant**: Is it naturally occurring?
**Environment**: Yes.
**Assistant**: Is it a mineral?
**Environment**: No.

**Assistant**: Is it a plant?
**Environment**: No.
**Assistant**: Is it a body of water?
**Environment**: Yes.
**Assistant**: Is it a sea?
**Environment**: No.
**Assistant**: Is it a lake?
**Environment**: No.
**Assistant**: Is it a river?
**Environment**: Yes.

Figure 9: Example of a multi-step interaction in the Twenty Questions task, where the agent narrows down possibilities through a sequence of yes/no questions.

**User:** I'm hosting a cocktail party and I need some cocktail recipes. Can you suggest some cocktail recipes that contain vodka as one of the ingredients? Also, provide me with a cake recipe from The Birthday Cake DB. Begin!

**Assistant:** "function": { "name": "search_by_ingredient_for_cocktails", "arguments": "{\n \"ingredient\": \"vodka\"\n}" }
**Tool:** {\"error\": \"\", \"response\": \"Vodka is a versatile and popular ingredient in cocktails, known for its neutral flavor that allows it to blend seamlessly with a variety of mixers. Some classic vodka cocktails include the Vodka Martini, typically made with vodka and dry vermouth; the Bloody Mary, a savory blend of vodka with tomato juice and various spices, often garnished with celery; and the Moscow Mule, which combines vodka with ginger beer and lime juice, traditionally served in a copper mug. Other popular options include the Cosmopolitan, which mixes vodka with triple sec, cranberry juice, and lime, and the Screwdriver, a simple yet refreshing combination of vodka and orange juice.\"}

**Assistant:** "function": { "name": "list_of_cakes_for_the_birthday_cake_db", "arguments": "{}"}
**Tool:** {\"error\": \"\", \"response\": \"[{\"id\":\"1\",\"title\":\"Raspberry and custard muffins\",\"difficulty\":\"Easy\",\"image\":\"https://apipics.s3.amazonaws.com/cakes_api/1.jpg\"},{\"id\":\"2\",\" title\":\"Lemon and blackberry stripe cake\",\"difficulty\":\"A challenge\",\"image\":\"https://apipics.s3.amazonaws.com/cakes_api/2.jpg\"},{\"id\":\"3\",\"title\":\"Paul Hollywood\\u2019s chocolate fudge cake\",\"difficulty\":\"Easy\",\"image\":\"https://apipics.s3.amazonaws.com/cakes_api/3.jpg\"},{\"id\":\"4\",\"title\":\"Lemon and strawberry meringue cake\",\"difficulty\":\" Easy\",\"image\":\"https://apipics.s3.amazonaws.com/cakes_api/4.jpg\"},{\"id\":\"5\",\"title\":\"Vegan chocolate cake\",\"difficulty\":\"Easy\",\"image\":\" https://apipics.s3.amazonaws.com/cakes_api/5.jpg\"},{\"id\":\"6\",\"title\":\"Spiced plum cake with swiss meringue frosting\",\"difficulty\":\"Easy\",\"image\":\" https://apipics.s3.amazonaws.com/cakes_api/6.jpg\"},{\"id\":\"7\",\"title\":\"Lemon and courgette cake with white chocolate cream cheese ...\",}

**Assistant:** "function" : { "name": "detailed_cake_recipe_by_id_for_the_birthday_cake_db", "arguments": "{\n \"is_id\": \"1\"\n}"}
**Tool:** "{\"error\": \"\", \"response\": {\"id\": \"1\", \"title\": \"Raspberry and custard muffins\", \"difficulty\": \"Easy\", \"portion\": \"Makes 12\", \"time\": \"Hands-on time 25 min. Oven time 18 min\", \"description\": \"Reminiscent of school traybakes with pink icing, these raspberry and custard muffins are given a suitably chic update \\u2013 simply by turning cupcakes upside down. Adding custard to the cake batter creates a moist crumb and another layer of nostalgia, that pairs brilliantly with the raspberry in the jam icing.\", \"ingredients\": [\"140g unsalted butter, melted\", \"3 medium free-range eggs\", \"125g custard\", \"1 tsp vanilla bean paste\", \"180g golden caster sugar\", \"180g self-raising flour\", \"4 tbsp seedless raspberry jam\", \"100g icing sugar\", \"Sprinkles to decorate\", \"12-hole muffin tin\"], \"method\": [{\"Step 1\": \"Heat the oven to 170\\u00b0C fan/gas 5. Brush around 20g of the melted butter into each hole of the muffin tin, ensuring some gets into every corner. Pop it in the fridge while you make the cake batter.\"}, {\"Step 2\": \"In...\", }"}

**Assistant:** "function": { "name": "Finish", "arguments": "{\n \"return_type\": \"give_answer\",\n \"final_answer\": \"Sure! Here are some cocktail recipes that contain vodka as one of the ingredients:\\n1. Vodka Martini\\n2. Bloody Mary\\n3. Moscow Mule\\n4. Cosmopolitan\\n5. Screwdriver\n\\nAnd here is a cake recipe from The Birthday Cake DB:\\n\\nRaspberry and custard muffins\\nDifficulty: Easy\\nPortion: Makes 12\\nTime: Hands-on time 25 min. Oven time 18 min\\nDescription: Reminiscent of school traybakes with pink icing, these raspberry and custard muffins are given a suitably chic update \\u2013 simply by turning cupcakes upside down. Adding custard to the cake batter creates a moist crumb and another layer of nostalgia, that pairs brilliantly with the raspberry in the jam icing.\\n\\nIngredients:\\n- 140g unsalted butter, melted\\n- 3 medium free-range eggs\\n- 125g custard\\n- 1 tsp vanilla bean paste\\n- 180g golden caster sugar\\n- 180g self-raising flour\\n- 4 tbsp seedless raspberry jam\\n- 100g icing sugar\\n- Sprinkles to decorate\\n- 12-hole muffin tin\\n\\nMethod:\\n1. Heat the oven to 170\\u00b0C fan/gas 5. Brush around 20g of the melted butter into each hole of the muffin tin, ensuring some gets into every corner. Pop it in the fridge while you make the cake batter.\\n2. In...\\n\\nEnjoy your cocktail party and the delicious cake!\"\n}"}

Figure 10: Example of a multi-step tool-based interaction in ToolBench, where the agent processes complex and lengthy tool responses across multiple steps to generate a final answer.

