# OpenReview forum: "StepTool: A Step-grained Reinforcement Learning Framework for Tool Learning in LLMs"
_ICLR.cc/2025/Conference — Submitted to ICLR 2025_

### Official Review · Reviewer_Nb46 · 2024-11-03

**Soundness:** 2
**Presentation:** 3
**Contribution:** 2
**Rating:** 5
**Confidence:** 4

**Summary:**

The paper introduces StepTool, a step-grained RL framework to improve tool learning in LLMs. StepTool enhances LLMs' ability to handle multi-step tasks by step-level reward shaping and optimization. Step-level reward shaping focuses on both the success of tool invocations and their contributions to task completion. Step-grained optimization is proposed based on policy gradient  to optimize the model in a interactive manner.

**Strengths:**

In general, it is a feasible RL framework for enhancing LLM, while the key lies in the design of the reward function for tool learning.

**Weaknesses:**

1. What is the difference between StepTool and PPO besides reward shaping in the experiments?
2. There exists many works [1,2,3,4] leveraging RL methods for language-argumented sequential decision-making problem. And the tool learning is a subset of language-argumented sequential decision-making problem. Thus authors should compare more carefully with these works instead of saying "Due to the lack of publicly available implementation details for some concurrent works, we choose two classic optimization methods: supervised fine-tuning (SFT) and PPO" in lines 346-347.
3. If StepTool has no core difference from the above works at the algorithm level, reward shaping alone does not seem to be enough to support an ICLR accepted article.

[1] Carta, Thomas, Clément Romac, Thomas Wolf, Sylvain Lamprier, Olivier Sigaud, and Pierre-Yves Oudeyer. "Grounding large language models in interactive environments with online reinforcement learning." In International Conference on Machine Learning, pp. 3676-3713. PMLR, 2023.

[2] Tan, Weihao, Wentao Zhang, Shanqi Liu, Longtao Zheng, Xinrun Wang, and Bo An. "True Knowledge Comes from Practice: Aligning Large Language Models with Embodied Environments via Reinforcement Learning." In The Twelfth International Conference on Learning Representations.

[3] Zhou, Yifei, Andrea Zanette, Jiayi Pan, Sergey Levine, and Aviral Kumar. "Archer: Training language model agents via hierarchical multi-turn rl." arXiv preprint arXiv:2402.19446 (2024).

[4] Wen, Muning, Ziyu Wan, Weinan Zhang, Jun Wang, and Ying Wen. "Reinforcing Language Agents via Policy Optimization with Action Decomposition." arXiv preprint arXiv:2405.15821 (2024).

**Questions:**

See Weaknesses

**Details Of Ethics Concerns:**

No ethics concerns.

---

> ### Author Response · Authors · 2024-11-20
> **Responses to Weakness 1**
>
> **W1: What is the difference between StepTool and PPO besides reward shaping in the experiments?**
>
> **A1:** The core differences lies in the utilization of training data: the "PPO" baseline, as a traditional RLHF implementation following [1], is not designed to handle multi-step data and therefore applies rewards only to the final step of a trajectory. In contrast, our framework supports step-grained reward integration across the multi-step trajectory, considering the interdependencies of multi-step actions in a trajectory.
>
> All other experimental configurations, including training data and reward annotations, were kept consistent between PPO and StepTool for fairness. Detailed experimental setups are available in the code repository to ensure transparency and reproducibility.
>
> We sincerely appreciate your valuable feedback and will clarify this distinction further in the experimental setup section of the revised paper.
>
> > [1] Long Ouyang, Jeffrey Wu, Xu Jiang, Diogo Almeida, Carroll Wainwright, Pamela Mishkin, Chong Zhang, Sandhini Agarwal, Katarina Slama, Alex Ray, et al. Training language models to follow instructions with human feedback. Advances in neural information processing systems, 35:27730–27744, 2022.

---

> ### Author Response · Authors · 2024-11-20
> **Responses to Weakness 2**
>
> **W2: There exists many works [1,2,3,4] leveraging RL methods for language-argumented sequential decision-making problem. And the tool learning is a subset of language-argumented sequential decision-making problem. Thus authors should compare more carefully with these works instead of saying "Due to the lack of publicly available implementation details for some concurrent works, we choose two classic optimization methods: supervised fine-tuning (SFT) and PPO" in lines 346-347.**
>
> **A2:** Thank you for your insightful comments regarding the connection between tool learning and language-augmented sequential decision-making problems. We deeply value the advancements in Language RL Agents [1-4], which highlight the potential of RL to align large language models with complex goals in interactive and dynamic environments. However, we would like to elaborate on two major challenges that arise when directly applying Language RL Agent approaches to tool learning:
>
> - Training paradigms: Language RL Agent tasks typically adopt an online training paradigm, where agents interact with the environment, receive explicit reward signals, and iteratively improve based on those signals [1-4]. In contrast, tool learning operates in a "semi-offline" context, more akin to RLHF setups [5]. In our framework, training involves improving the model’s performance on pre-generated (sampled) multi-step trajectories in a round-based process, where training data is iteratively collected. These offline trajectories include tool invocation responses but lack explicit or real-time reward signals. This difference in paradigm makes a direct transfer of Language RL Agent methods to tool learning nontrivial.
> - Task Characteristics and Action Space: Language RL Agents often operate within relatively constrained environments, such as Overcooked, VirtualHome [2,4], or BabyAI-Text [1], where the actions usually can be grounded within a relatively limited action space. The overlap between training and evaluation actions in such tasks is substantial, which is typical for RL-based agents. Tool learning, however, features a significantly larger action space, encompassing thousands of APIs and parameter configurations. This scale introduces challenges in grounding and action enumeration. Moreover, in tool learning, it is common for evaluation tasks to involve tools that were not seen during training, as described in the ToolBench task design [6]. This introduces challenges related to generalization, grounding, and handling unseen tools.
>
> Given these challenges, directly applying Language RL Agent methods to tool learning could result in inefficiencies or performance degradation. While exploring this transferability is an exciting direction for future work, it would require significant adaptation to align with the unique characteristics of tool learning environments.
>
> We appreciate your thoughtful feedback and will revise the phrasing in the manuscript to better reflect the scope and context of our work. Furthermore, we believe that investigating the applicability of Language RL Agent methods to tool learning could open valuable avenues for future research and significantly enrich the field.
>
>
>
> > [1] Carta, Thomas, Clément Romac, Thomas Wolf, Sylvain Lamprier, Olivier Sigaud, and Pierre-Yves Oudeyer. "Grounding large language models in interactive environments with online reinforcement learning." In International Conference on Machine Learning, pp. 3676-3713. PMLR, 2023.
> >
> > [2] Tan, Weihao, Wentao Zhang, Shanqi Liu, Longtao Zheng, Xinrun Wang, and Bo An. "True Knowledge Comes from Practice: Aligning Large Language Models with Embodied Environments via Reinforcement Learning." In The Twelfth International Conference on Learning Representations.
> >
> > [3] Zhou, Yifei, Andrea Zanette, Jiayi Pan, Sergey Levine, and Aviral Kumar. "Archer: Training language model agents via hierarchical multi-turn rl." arXiv preprint arXiv:2402.19446 (2024).
> >
> > [4] Wen, Muning, Ziyu Wan, Weinan Zhang, Jun Wang, and Ying Wen. "Reinforcing Language Agents via Policy Optimization with Action Decomposition." arXiv preprint arXiv:2405.15821 (2024).
> >
> > [5] Long Ouyang, Jeffrey Wu, Xu Jiang, Diogo Almeida, Carroll Wainwright, Pamela Mishkin, Chong Zhang, Sandhini Agarwal, Katarina Slama, Alex Ray, et al. Training language models to follow instructions with human feedback. Advances in neural information processing systems, 35:27730–27744, 2022.
> >
> > [6] Yujia Qin, Shihao Liang, Yining Ye, Kunlun Zhu, Lan Yan, Yaxi Lu, Yankai Lin, Xin Cong, Xiangru Tang, Bill Qian, et al. Toolllm: Facilitating large language models to master 16000+ real-world apis. arXiv preprint arXiv:2307.16789, 2023.

---

> ### Author Response · Authors · 2024-11-20
> **Responses to Weakness 3**
>
> **W3: If StepTool has no core difference from the above works at the algorithm level, reward shaping alone does not seem to be enough to support an ICLR accepted article.**
>
> **A3:** Thank you for your thoughtful feedback. We would like to take this opportunity to clarify the novelty and contributions of our work.
>
> Tool learning is an emerging scenario to extend the capabilities of large language models (LLMs) by enabling them to utilize external tools (APIs). Most existing approaches focus on data construction and supervised fine-tuning (SFT) methods to improve tool-use capabilities [1-4].
>
> Our work is the first to explore tool learning from a reinforcement learning (RL) perspective, introducing a reward shaping mechanism to supervise intermediate steps and an optimization framework compatible with various policy-gradient-based RL algorithms. Unlike traditional RLHF methods [5], StepTool provides process-level supervision for intermediate steps and explicitly considers the interdependencies between multi-step actions in a trajectory during optimization.
>
> **Broader Implications of StepTool**: This work provides a fresh perspective distinct from the dominant SFT-based approaches [1-4], demonstrating the potential of RL as a complementary method for optimizing tool-use tasks. As a foundational contribution, StepTool establishes a general framework that can inspire future innovations in RL techniques tailored to tool learning.
>
> Thank you again for your insightful feedback. We view this work as a meaningful step toward broadening the application of RL methodologies in tool learning, and we look forward to contributing to continued advancements in this evolving field.
>
> > [1] Ibrahim Abdelaziz, Kinjal Basu, Mayank Agarwal, Sadhana Kumaravel, Matthew Stallone, Rameswar Panda, Yara Rizk, GP Bhargav, Maxwell Crouse, Chulaka Gunasekara, et al. Granite-function calling model: Introducing function calling abilities via multi-task learning of granular tasks. arXiv preprint arXiv:2407.00121, 2024.
> >
> > [2] Zuxin Liu, Thai Hoang, Jianguo Zhang, Ming Zhu, Tian Lan, Shirley Kokane, Juntao Tan, Weiran Yao, Zhiwei Liu, Yihao Feng, et al. Apigen: Automated pipeline for generating verifiable and diverse function-calling datasets. arXiv preprint arXiv:2406.18518, 2024.
> >
> > [3] Yujia Qin, Shihao Liang, Yining Ye, Kunlun Zhu, Lan Yan, Yaxi Lu, Yankai Lin, Xin Cong, Xiangru Tang, Bill Qian, et al. Toolllm: Facilitating large language models to master 16000+ real-world apis. arXiv preprint arXiv:2307.16789, 2023.
> >
> > [4] Shishir G Patil, Tianjun Zhang, Xin Wang, and Joseph E Gonzalez. Gorilla: Large language model connected with massive apis. arXiv preprint arXiv:2305.15334, 2023.
> >
> > [5] Long Ouyang, Jeffrey Wu, Xu Jiang, Diogo Almeida, Carroll Wainwright, Pamela Mishkin, Chong Zhang, Sandhini Agarwal, Katarina Slama, Alex Ray, et al. Training language models to follow instructions with human feedback. Advances in neural information processing systems, 35:27730–27744, 2022.

---

> > ### Comment · Reviewer_Nb46 · 2024-11-23
> > **Still have some concerns about the contributions outside of previous works**
> >
> > Thanks for authors; responses, but I still have some concerns about the contributions outside of previous works [1-4]:
> > 1. The large action space problem mentioned by the author has also been solved in previous works[3,4].
> > 2. Tasks involving actions (corresponding to tool-using here) that were not seen during training are considered in [2,4] as well.
> > 3. The author said their trajectories are offline. However, I don't understand how an on-policy PPO algorithm can be used with offline data. (Did authors train a reward model?)
> >
> > Therefore, when considering the multi-turn tool learning as a subset of language-argument sequential decision-making, what is the irreplaceable contribution of this work besides reward shaping?
> >
> > [1] Carta, Thomas, Clément Romac, Thomas Wolf, Sylvain Lamprier, Olivier Sigaud, and Pierre-Yves Oudeyer. "Grounding large language models in interactive environments with online reinforcement learning." In International Conference on Machine Learning, pp. 3676-3713. PMLR, 2023.
> >
> > [2] Tan, Weihao, Wentao Zhang, Shanqi Liu, Longtao Zheng, Xinrun Wang, and Bo An. "True Knowledge Comes from Practice: Aligning Large Language Models with Embodied Environments via Reinforcement Learning." In The Twelfth International Conference on Learning Representations.
> >
> > [3] Zhou, Yifei, Andrea Zanette, Jiayi Pan, Sergey Levine, and Aviral Kumar. "Archer: Training language model agents via hierarchical multi-turn rl." arXiv preprint arXiv:2402.19446 (2024).
> >
> > [4] Wen, Muning, Ziyu Wan, Weinan Zhang, Jun Wang, and Ying Wen. "Reinforcing Language Agents via Policy Optimization with Action Decomposition." arXiv preprint arXiv:2405.15821 (2024).

---

> > > ### Author Response · Authors · 2024-11-24
> > > **Responses to Contribution and Training Paradigm**
> > >
> > > Thank you very much for your thoughtful feedback, as well as for highlighting references addressing large action space and generalization. These provide valuable directions for our future work. Regarding the contributions of this work and your third question about our training paradigm, we would like to provide further clarification.
> > >
> > > **Contribution**: Tool Learning is a significant and emerging task aimed at extending the capability boundaries of LLMs. Our work is **the first to model Tool Learning as a sequential decision-making problem** and to explore its optimization through RL methods. By introducing this novel problem modeling, this work has opened the door to discussions about how other language-augmented sequential decision-making works might be adapted, which would be **an important direction for our future efforts**. As the first attempt to introduce this modeling and solution, we adopted relatively straightforward and simple approaches to address the challenges, aiming to **provide a foundational framework and a new perspective for this field**. While this work focuses on framing the problem and providing process supervision insights, future research will explore more sophisticated algorithmic innovations to enhance performance.
> > >
> > > **Training Paradigm**: Regarding the **"semi-offline"** training setup mentioned in our response, it refers to an iterative process: first, we collect (sample)  a set of trajectories under the current model, then train the model using RL methods on this collected data. After updating the model, we fix it and repeat the process—collecting (sampling) new trajectories, during which the reward model provides feedback. **This iterative approach is not fully online (as RL updates are not performed simultaneously with environment interaction) nor fully offline (as the data is periodically refreshed after each training round)**. This setup is **a standard practice in RLHF**, as described in the work by Long Ouyang et al. ("Training language models to follow instructions with human feedback," NeurIPS 2022). We will clarify this aspect in the manuscript to avoid confusion.
> > >
> > > Thank you again for your detailed comments, which have greatly helped us refine the positioning and explanation of our work. Tool Learning itself is a very new and evolving field that requires significant exploration and innovation. Our work represents an important first step toward providing new problem modeling and optimization directions for this task. We hope this response helps you better understand the contributions and novelty of our work within the context of the nascent Tool Learning field.

---

> > > > ### Comment · Reviewer_Nb46 · 2024-11-26
> > > > **Thanks for authors' response**
> > > >
> > > > Thanks for the response. I agree this work should be the first to model (multi-turn) tool learning as a sequential decision-making problem.
> > > >
> > > > However, I can't agree with the statement "This iterative approach is not fully online (as RL updates are not performed simultaneously with environment interaction) nor fully offline (as the data is periodically refreshed after each training round)". According to the authors' description " first, we collect (sample) a set of trajectories under the current model, then train the model using RL methods on this collected data. After updating the model, we fix it and repeat the process—collecting (sampling) new trajectories, during which the reward model provides feedback", it is a very classical **online on-policy** RL paradigm - collecting a batch of trajectories with current policies, updating the policies with the batch, fixing the policies and collecting the next batch,... - That is what I concern this training process is similar to previous works [1-4].

---

> > > > > ### Author Response · Authors · 2024-11-26
> > > > > **Thanks for your responses**
> > > > >
> > > > > Thank you very much for your thoughtful feedback and for pointing out the issue with our previous explanation. Upon carefully reviewing your comment and revisiting the references [1-4], we realize that our earlier statement was inaccurate. The training process indeed follows the classical online on-policy reinforcement learning paradigm, as you described.
> > > > >
> > > > > We sincerely apologize for the oversight in our description and for any confusion it may have caused. We are currently in the process of reproducing these works and will include them as baseline comparisons in our revised manuscript to provide a clearer context.
> > > > >
> > > > > Your review has provided us with a valuable perspective, helping us reassess the limitations of our work. Even if this submission is not accepted, your feedback will serve as an important guide for improving this work going forward.
> > > > >
> > > > > Thank you again for your time and consideration.

---

> > > > > > ### Comment · Reviewer_Nb46 · 2024-11-26
> > > > > > **Thanks for detailed responses.**
> > > > > >
> > > > > > Thanks for the detailed responses. These clarifications have addressed some of my concerns.
> > > > > >
> > > > > > Taking all these 4 works as baselines may not be necessary, since there should be more than these four concrete RL algorithms for multi-step textual tasks and they are just some examples. Instead, authors need **very careful** discussion about the connection and differences between StepTool and them, as well as other potential works that apply RL for multi-step textual tasks.
> > > > > >
> > > > > > I am willing to see this paper be more clear and solid in a later version and raise my score to 5 in advance.

---

> > > > > > > ### Author Response · Authors · 2024-11-28
> > > > > > > **Responses: Clarifying the Relationship Between StepTool and RL Approaches for Multi-Step Textual Tasks**
> > > > > > >
> > > > > > > We sincerely appreciate your thoughtful feedback and detailed questions regarding the relationship between StepTool and other works that apply RL for multi-step textual tasks. Below, we clarify the connections and differences in terms of both scenarios and methods, addressing your concerns.
> > > > > > >
> > > > > > > ### **Scenario Perspective**
> > > > > > >
> > > > > > > **Connection**: Similar to prior works, our approach operates within text-based environments where the agent interacts to solve multi-step tasks.
> > > > > > >
> > > > > > > **Differences**: Tasks like *Twenty Questions*, *Guess My City*, and *Webshop* involve relatively short actions and observations, where predefined goals allow for simple reward mechanisms. For example, in *Twenty Questions*, the ground truth is often a single object or word, making it easy to evaluate the agent's performance at each step. In contrast, tool learning presents unique challenges. The actions of tool invocation require adaptive parameters in JSON format, while the observations (responses) from the environment are lengthy and complex. As illustrated in the examples below, consider a query like *"Find cocktail recipes containing vodka and suggest a cake recipe from The Birthday Cake DB."* The intermediate tool responses often return large, detailed outputs, such as lists of cocktail recipes or multi-step cake preparation instructions.
> > > > > > >
> > > > > > > Furthermore, the assistant’s final answer, which addresses the user’s query, is often complex and open-ended. Determining rewards—both intermediate and final—requires nuanced evaluation and cannot rely on straightforward ground-truth matching.
> > > > > > >
> > > > > > >
> > > > > > > ### **Method Perspective**
> > > > > > >
> > > > > > > From a methodological perspective, StepTool and prior works share the use of RL to model decision-making and optimize performance. Among the many RL algorithms that could be applied to multi-step textual tasks, works [1-4] are particularly representative, and we use them as examples to highlight the connections and differences.
> > > > > > >
> > > > > > > - **Comparison with Works [1-2]**: As noted in work [4], works [1-2] manually constrain the action space to a restricted subset, focusing on optimizing actions as a whole. Probabilities are estimated for potential actions by combining the conditional probabilities of their composing tokens. This approach suits structured environments but does not scale to the diverse and expansive action space in tool learning.
> > > > > > > - **Comparison with Works [3-4]**: These works share similarities with StepTool, emphasizing token-level optimization and credit assignments. However, they rely on action-level models, introducing Q-functions $Q(s,a)$and value functions $V(s)$, to estimate token-level advantages. These models propagate intra-action influences at the action level, which are then distributed to the token level. This reliance on auxiliary models like $Q$ and $V$ makes optimization dependent on their accuracy and training stability. In contrast, StepTool directly computes token-level advantages, capturing intra-action and inter-action influences without requiring intermediate action-level estimations. This design avoids potential inaccuracies in Q and V models, providing a more robust and efficient optimization framework for tool learning.
> > > > > > >
> > > > > > > We are currently implementing works [3-4] as baselines and conducting experiments to validate our contributions further. Due to time constraints, the results are not yet complete, but we will release them promptly once finalized. Additionally, we will incorporate the detailed comparisons outlined above into the revised manuscript for clarity and completeness. Thank you again for your valuable feedback, which has been instrumental in refining our work.
> > > > > > >
> > > > > > > > [1] Carta, Thomas, Clément Romac, Thomas Wolf, Sylvain Lamprier, Olivier Sigaud, and Pierre-Yves Oudeyer. "Grounding large language models in interactive environments with online reinforcement learning." In International Conference on Machine Learning, pp. 3676-3713. PMLR, 2023.
> > > > > > > >
> > > > > > > > [2] Tan, Weihao, Wentao Zhang, Shanqi Liu, Longtao Zheng, Xinrun Wang, and Bo An. "True Knowledge Comes from Practice: Aligning Large Language Models with Embodied Environments via Reinforcement Learning." In The Twelfth International Conference on Learning Representations.
> > > > > > > >
> > > > > > > > [3] Zhou, Yifei, Andrea Zanette, Jiayi Pan, Sergey Levine, and Aviral Kumar. "Archer: Training language model agents via hierarchical multi-turn rl." arXiv preprint arXiv:2402.19446 (2024).
> > > > > > > >
> > > > > > > > [4] Wen, Muning, Ziyu Wan, Weinan Zhang, Jun Wang, and Ying Wen. "Reinforcing Language Agents via Policy Optimization with Action Decomposition." arXiv preprint arXiv:2405.15821 (2024).

---

> > > > > > > ### Author Response · Authors · 2024-11-28
> > > > > > > **Examples of TwentyQuestions and ToolBench**
> > > > > > >
> > > > > > > > Examples of *Twenty Questions*
> > > > > > > ```json
> > > > > > > {
> > > > > > >     "lines": [
> > > > > > >         "Is it an animal? No.",
> > > > > > >         "Is it an inanimate object? Yes.",
> > > > > > >         "Is it man-made? No.",
> > > > > > >         "Is it naturally occurring? Yes.",
> > > > > > >         "Is it a mineral? No.",
> > > > > > >         "Is it a plant? No.",
> > > > > > >         "Is it a body of water? Yes.",
> > > > > > >         "Is it a sea? No.",
> > > > > > >         "Is it a lake? No.",
> > > > > > >         "Is it a river? Yes."
> > > > > > >     ],
> > > > > > >     "correct": true,
> > > > > > >     "word": [
> > > > > > >         "River"
> > > > > > >     ]
> > > > > > > }
> > > > > > > ```
> > > > > > > > Examples of *Toolbench*
> > > > > > > ```json
> > > > > > > "convs": [
> > > > > > >     {
> > > > > > >         "role": "user",
> > > > > > >         "content": "\nI'm hosting a cocktail party and I need some cocktail recipes. Can you suggest some cocktail recipes that contain vodka as one of the ingredients? Also, provide me with a cake recipe from The Birthday Cake DB.\nBegin!\n"
> > > > > > >     },
> > > > > > >     {
> > > > > > >         "role": "assistant",
> > > > > > >         "function": {
> > > > > > >             "name": "search_by_ingredient_for_cocktails",
> > > > > > >             "arguments": "{\n  \"ingredient\": \"vodka\"\n}"
> > > > > > >         }
> > > > > > >     },
> > > > > > >     {
> > > > > > >         "role": "tool",
> > > > > > >         "name": "search_by_ingredient_for_cocktails",
> > > > > > >         "content": "{\"error\": \"\", \"response\": \"Vodka is a versatile and popular ingredient in cocktails, known for its neutral flavor that allows it to blend seamlessly with a variety of mixers. Some classic vodka cocktails include the Vodka Martini, typically made with vodka and dry vermouth; the Bloody Mary, a savory blend of vodka with tomato juice and various spices, often garnished with celery; and the Moscow Mule, which combines vodka with ginger beer and lime juice, traditionally served in a copper mug. Other popular options include the Cosmopolitan, which mixes vodka with triple sec, cranberry juice, and lime, and the Screwdriver, a simple yet refreshing combination of vodka and orange juice.\"}",
> > > > > > >     },
> > > > > > >     {
> > > > > > >         "role": "assistant",
> > > > > > >         "function": {
> > > > > > >             "name": "list_of_cakes_for_the_birthday_cake_db",
> > > > > > >             "arguments": "{}"
> > > > > > >         }
> > > > > > >     },
> > > > > > >     {
> > > > > > >         "role": "tool",
> > > > > > >         "name": "list_of_cakes_for_the_birthday_cake_db",
> > > > > > >         "content": "{\"error\": \"\", \"response\": \"[{"id":"1","title":"Raspberry and custard muffins","difficulty":"Easy","image":"https://apipics.s3.amazonaws.com/cakes_api/1.jpg"},{"id":"2","title":"Lemon and blackberry stripe cake","difficulty":"A challenge","image":"https://apipics.s3.amazonaws.com/cakes_api/2.jpg"},{"id":"3","title":"Paul Hollywood\\u2019s chocolate fudge cake","difficulty":"Easy","image":"https://apipics.s3.amazonaws.com/cakes_api/3.jpg"},{"id":"4","title":"Lemon and strawberry meringue cake","difficulty":"Easy","image":"https://apipics.s3.amazonaws.com/cakes_api/4.jpg"},{"id":"5","title":"Vegan chocolate cake","difficulty":"Easy","image":"https://apipics.s3.amazonaws.com/cakes_api/5.jpg"},{"id":"6","title":"Spiced plum cake with swiss meringue frosting","difficulty":"Easy","image":"https://apipics.s3.amazonaws.com/cakes_api/6.jpg"},{"id":"7","title":"Lemon and courgette cake with white chocolate cream cheese ...",
> > > > > > >     },
> > > > > > >     {
> > > > > > >         "role": "assistant",
> > > > > > >         "function": {
> > > > > > >             "name": "detailed_cake_recipe_by_id_for_the_birthday_cake_db",
> > > > > > >             "arguments": "{\n  \"is_id\": \"1\"\n}"
> > > > > > >         }
> > > > > > >     },
> > > > > > >     {
> > > > > > >         "role": "tool",
> > > > > > >         "name": "detailed_cake_recipe_by_id_for_the_birthday_cake_db",
> > > > > > >         "content": "{\"error\": \"\", \"response\": {\"id\": \"1\", \"title\": \"Raspberry and custard muffins\", \"difficulty\": \"Easy\", \"portion\": \"Makes 12\", \"time\": \"Hands-on time 25 min. Oven time 18 min\", \"description\": \"Reminiscent of school traybakes with pink icing, these raspberry and custard muffins are given a suitably chic update \\u2013 simply by turning cupcakes upside down. Adding custard to the cake batter creates a moist crumb and another layer of nostalgia, that pairs brilliantly with the raspberry in the jam icing.\", \"ingredients\": [\"140g unsalted butter, melted\", \"3 medium free-range eggs\", \"125g custard\", \"1 tsp vanilla bean paste\", \"180g golden caster sugar\", \"180g self-raising flour\", \"4 tbsp seedless raspberry jam\", \"100g icing sugar\", \"Sprinkles to decorate\", \"12-hole muffin tin\"], \"method\": [{\"Step 1\": \"Heat the oven to 170\\u00b0C fan/gas 5. Brush around 20g of the melted butter into each hole of the muffin tin, ensuring some gets into every corner. Pop it in the fridge while you make the cake batter.\"}, {\"Step 2\": \"In...",
> > > > > > >     },
> > > > > > >     ...
> > > > > > > }

---

> > > > > > > > ### Author Response · Authors · 2024-11-28
> > > > > > > > **Examples of ToolBench (Continued)**
> > > > > > > >
> > > > > > > > > Examples of *Toolbench* (Continued)
> > > > > > > > ```json
> > > > > > > > "convs": [
> > > > > > > >     ...
> > > > > > > >     {
> > > > > > > >         "role": "assistant",
> > > > > > > >         "function": {
> > > > > > > >             "name": "Finish",
> > > > > > > >             "arguments": "{\n  \"return_type\": \"give_answer\",\n  \"final_answer\": \"Sure! Here are some cocktail recipes that contain vodka as one of the ingredients:\\n1. Vodka Martini\\n2. Bloody Mary\\n3. Moscow Mule\\n4. Cosmopolitan\\n5. Screwdriver\\n\\nAnd here is a cake recipe from The Birthday Cake DB:\\n\\nRaspberry and custard muffins\\nDifficulty: Easy\\nPortion: Makes 12\\nTime: Hands-on time 25 min. Oven time 18 min\\nDescription: Reminiscent of school traybakes with pink icing, these raspberry and custard muffins are given a suitably chic update \u2013 simply by turning cupcakes upside down. Adding custard to the cake batter creates a moist crumb and another layer of nostalgia, that pairs brilliantly with the raspberry in the jam icing.\\n\\nIngredients:\\n- 140g unsalted butter, melted\\n- 3 medium free-range eggs\\n- 125g custard\\n- 1 tsp vanilla bean paste\\n- 180g golden caster sugar\\n- 180g self-raising flour\\n- 4 tbsp seedless raspberry jam\\n- 100g icing sugar\\n- Sprinkles to decorate\\n- 12-hole muffin tin\\n\\nMethod:\\n1. Heat the oven to 170\u00b0C fan/gas 5. Brush around 20g of the melted butter into each hole of the muffin tin, ensuring some gets into every corner. Pop it in the fridge while you make the cake batter.\\n2. In...\\n\\nEnjoy your cocktail party and the delicious cake!\"\n}"
> > > > > > > >         }
> > > > > > > >     }
> > > > > > > > ]
> > > > > > > > ```

---

### Official Review · Reviewer_UjDC · 2024-11-03

**Soundness:** 2
**Presentation:** 3
**Contribution:** 2
**Rating:** 5
**Confidence:** 3

**Summary:**

The paper introduces the concept of step-grained reward shaping to enhance the reinforcement learning process. The proposed method focuses on the design of auxiliary rewards aimed at facilitating task completion. However, the reliance on these manually designed rewards raises questions about potential conflicts with the ground truth reward and issues related to reward hacking. A deeper discussion on these points is necessary, along with clarification on how contribution metrics

**Strengths:**

* The paper addresses an challenge for LLM tool using by proposing auxiliary rewards to guide the learning process, which could improve the efficiency of solving complex tasks for LLMs.

**Weaknesses:**

* The reliance on auxiliary rewards introduces the risk of conflicts with the ground truth reward, which is a significant concern that the authors need to address more thoroughly. The potential for reward hacking must be discussed in detail.
* The paper lacks empirical analysis on the average number of tools called during task execution and how the complexity of tasks influences the length of action sequences. An exploration of whether long sequences stem from planning or task decomposition issues would enhance the understanding of the framework's practical applicability.

**Questions:**

1. Could the authors elaborate on how the auxiliary rewards might conflict with the ground truth reward and the implications of such conflicts?
2. How exactly are the contribution metric/reward trained?
3. What is the average number of tools utilized per task, and how does this correlate with the difficulty of the tasks? If the step length is excessive, could this indicate deficiencies in planning or task decomposition?

Minor Comments:
* Line 176 contains an inaccuracy: “advantage which compares the expected return of a given action to the average return for that state.” The state value refers to the expected overall value of a state, not the average return.

---

> ### Author Response · Authors · 2024-11-20
> **Responses to Question 1-2**
>
> **Q1: Could the authors elaborate on how the auxiliary rewards might conflict with the ground truth reward and the implications of such conflicts?**
>
> **A1:** Thank you for raising this question. We are not sure whether the "conflicts with the ground truth reward" refers to discrepancies between manually designed rewards and environment-provided rewards, or potential estimation errors during reward annotation.
>
> If the concern involves **conflicts between manually designed rewards and environment-provided rewards**, it is worth noting that tool learning environments often lack explicit or ground-truth reward signals. In such cases, our work is the first to provide a practical solution by defining auxiliary rewards to approximate the intended outcomes of tool usage. However, since ground truth rewards are unavailable, such conflicts cannot currently be detected.
>
> If the concern relates to the **potential reward annotation errors**, these could arise during automated or manual labeling of auxiliary rewards or result from reward hacking. To improve the robustness of reward annotations and mitigate such errors, several strategies could be employed:
>
> 1. Reward Model Ensembles: Using multiple reward models and averaging their outputs can reduce the impact of individual model biases.
> 2. Multiple Annotations: Aggregating annotations from diverse sources can provide a more balanced reward signal.
> 3. Iterative Refinement: Refining reward annotations iteratively based on experimental results can improve alignment with the task's ultimate objectives.
>
> We appreciate your feedback on this critical issue and hope this explanation addresses your question. If further clarification or discussion is required, we would be happy to elaborate and include these points in the revised paper.
>
>
>
> **Q2: How exactly are the contribution metric/reward trained?**
>
> **A2:** The values of our rewards are obtained from both rule-based judgments and GPT-4-based annotations, without training. Reward shaping in our framework includes three main components/functions, *SuccCalling*, *Contribution* and *IsSolved*. Specifically, in our experiments:
>
> - The values of *SuccCalling* were determined through a combination of rule-based and GPT-4-based annotations. Rules were employed to evaluate the format of the tool invocation (e.g., correctness of tool names and parameters). For invocations meeting the format requirements, GPT-4 was used to assess the actual success based on the tool response (e.g., a response like “Since the search parameter is empty, please provide the name of an author…” indicates a failed invocation due to incorrect arguments).
> - The values of *Contribution* and *IsSolved* were obtained exclusively through automated annotations by GPT-4, which leverages its semantic analysis capabilities.
>
> We sincerely appreciate your feedback and will incorporate these explanations into the "Step-Grained Reward Acquisition" section of the revised paper.

---

> ### Author Response · Authors · 2024-11-20
> **Responses to Question 3**
>
> **Q3: What is the average number of tools utilized per task, and how does this correlate with the difficulty of the tasks? If the step length is excessive, could this indicate deficiencies in planning or task decomposition?**
>
> **A3:** Thank you for highlighting this area for deeper analysis. First, let us provide a detailed description of the task setup in the StableToolBench benchmark [2]. In StableToolBench, each task typically requires 2-3 API calls for successful completion, accounting for 88.5% of the tasks (68.6% requiring 2 APIs and 19.9% requiring 3 APIs).  Task difficulty is primarily influenced by factors such as tool diversity (e.g., intra-tool, intra-category, or intra-collection variations) and whether the required tools are included in the training set. More details on task difficulty design can be found in the ToolBench paper [1].
>
> Regarding the model's performance on the average number of tools utilized, we calculated the average tool invocations per task across subsets and model configurations. The results are summarized in the following table:
>
>
> | BaseModel | Strategy | Method       | I1 Ins. | I1 Cat. | I1 Tool | I2 Ins. | I2 Cat. | I3 Ins. |
> | :-------- | :------- | :----------- | ------: | ------: | ------: | ------: | ------: | ------: |
> | ToolLlama | CoT      | /          | 2.20245 |  2.1634 | 2.36076 | 2.46226 | 2.55645 | 2.57377 |
> | ToolLlama | CoT      | RLHF-PPO     | 2.20245 | 2.09804 | 2.29114 | 2.40566 | 2.49194 | 2.54098 |
> | ToolLlama | CoT      | StepTool | 2.25153 | 2.14379 | 2.32278 | 2.49057 | 2.50806 |  2.4918 |
> | ToolLlama | DFSDT    | /          | 2.15951 | 2.09804 | 2.20253 | 2.38679 | 2.51613 | 2.54098 |
> | ToolLlama | DFSDT    | RLHF-PPO     | 2.17791 |  2.0719 | 2.23418 | 2.32075 | 2.45161 |  2.4918 |
> | ToolLlama | DFSDT    | StepTool | 2.20245 | 2.10458 | 2.25949 | 2.45283 | 2.50806 | 2.39344 |
> | Llama3.1  | CoT      | SFT          | 2.17791 | 2.14379 | 2.17089 | 2.39623 | 2.41935 |  2.4918 |
> | Llama3.1  | CoT      | RLHF-PPO     | 2.15337 | 2.11765 | 2.21519 | 2.41509 | 2.39516 | 2.55738 |
> | Llama3.1  | CoT      | StepTool | 2.16564 | 2.04575 |  2.1962 | 2.42453 | 2.32258 | 2.37705 |
> | Llama3.1  | DFSDT    | SFT          | 2.17178 | 2.14379 |  2.1519 | 2.41509 |  2.3871 | 2.42623 |
> | Llama3.1  | DFSDT    | RLHF-PPO     | 2.16564 | 2.11765 | 2.21519 | 2.42453 | 2.37903 | 2.47541 |
> | Llama3.1  | DFSDT    | StepTool | 2.19018 | 2.01307 | 2.12025 | 2.42453 | 2.29839 | 2.39344 |
> | Qwen2     | CoT      | SFT          | 2.22086 | 2.08497 | 2.21519 | 2.43396 | 2.42742 | 2.57377 |
> | Qwen2     | CoT      | RLHF-PPO     | 2.21472 | 2.16993 | 2.25191 |  2.4434 | 2.42742 | 2.55738 |
> | Qwen2     | CoT      | StepTool | 2.19018 |  2.1634 | 2.22152 |  2.4717 | 2.41935 | 2.59016 |
> | Qwen2     | DFSDT    | SFT          | 2.19632 | 2.12418 | 2.24051 | 2.40566 | 2.43548 |  2.5082 |
> | Qwen2     | DFSDT    | RLHF-PPO     | 2.20245 | 2.15033 | 2.25316 | 2.42453 | 2.43548 | 2.47541 |
> | Qwen2     | DFSDT    | StepTool | 2.18405 | 2.09804 | 2.18987 | 2.40566 | 2.41129 | 2.59016 |
>
> From the results, we observe the following key points:
>
> - The average number of tool invocations per task shows minimal variation (<0.1 tool calls per task) across models under the same experimental settings. This observation is highly related to the task design in the benchmark, as mentioned above.
> - The number of tool invocations does not directly correlate with model performance. This is because fewer tool invocations may indicate omitted critical API calls, while more tools invocations could reflect redundant steps.
>
> In summary, we believe that step length could serve as a valuable metric for evaluating a model's planning and task decomposition capabilities. However, this aspect cannot be fully assessed within the constraints of this benchmark.
>
> We appreciate your insightful feedback and will include these results and discussions in the Appendix of the revised paper. Additionally, we are grateful for your correction regarding our statements on Line 176 and will revise it in future versions of the paper.
>
> [1] Yujia Qin, Shihao Liang, Yining Ye, Kunlun Zhu, Lan Yan, Yaxi Lu, Yankai Lin, Xin Cong, Xiangru Tang, Bill Qian, et al. Toolllm: Facilitating large language models to master 16000+ real-world apis. arXiv preprint arXiv:2307.16789, 2023.
>
> [2] Zhicheng Guo, Sijie Cheng, Hao Wang, Shihao Liang, Yujia Qin, Peng Li, Zhiyuan Liu, Maosong Sun, and Yang Liu. Stabletoolbench: Towards stable large-scale benchmarking on tool learning of large language models. arXiv preprint arXiv:2403.07714, 2024.

---

> > ### Comment · Reviewer_UjDC · 2024-11-26
> >
> > Thanks for detailed responses. These clarifications have addressed some of my concerns, and I have raised my score.
> >
> > However, I still think that rule-based judgments and GPT-4-based annotations might require careful design and weight adjustment to be truly effective. The potential conflicts between auxiliary rewards, as well as the risk of reward hacking could have further discussion.
> >
> > I also agree with reviewer Nb46 that the main contribution of your work is the fine-grained reward design and shaping for the one-turn tool learning task.

---

> > > ### Author Response · Authors · 2024-11-28
> > > **Thanks for your responses**
> > >
> > > Thank you for your thoughtful feedback and for raising your score—your insights have been immensely helpful in refining our work. We greatly appreciate your acknowledgment of the detailed responses provided.
> > >
> > > Regarding your point on the potential conflicts between auxiliary rewards and the risk of reward hacking, we recognize the significance of these issues and agree that they warrant careful consideration. To address this, we have added further discussions to the appendix of the revised paper, elaborating on how auxiliary rewards are designed and balanced, as well as the measures implemented to mitigate reward hacking. Your comments have greatly helped us highlight these aspects, which will undoubtedly guide future iterations of this work.
> > >
> > > We would also like to take this opportunity to further clarify the novelty and contributions of our study. Tool Learning is a significant and emerging task aimed at extending the capability boundaries of LLMs. Our work is **the first to model Tool Learning as a sequential decision-making problem** and to explore its optimization through RL methods. By introducing this novel problem modeling, this work has opened the door to discussions about how other language-augmented sequential decision-making works might be adapted, which would be **an important direction for our future efforts**. As the first attempt to introduce this modeling and solution, we adopted relatively straightforward and simple approaches to address the challenges, aiming to **provide a foundational framework and a new perspective for this field**. While this work focuses on framing the problem and providing process supervision insights, future research will explore more sophisticated algorithmic innovations to enhance performance.
> > >
> > > Your review has provided us with a valuable perspective, helping us reassess the limitations of our work and identify critical areas for improvement. Your feedback will serve as an important guide for enhancing this work and advancing this field. We remain open to any additional suggestions you may have.

---

### Official Review · Reviewer_tiFx · 2024-11-04

**Soundness:** 3
**Presentation:** 2
**Contribution:** 3
**Rating:** 6
**Confidence:** 3

**Summary:**

This paper proposes to apply (multi-step) RL for optimizing LLM tool use capabilities. To instantiate this, they design a step-level (aka process-level) reward consisting of terms that reward the correctness of the tool call, the relevance of the tool call to the successful completion, and the final success reward. Experimental results show small to medium improvements over non-RL baselines.

**Strengths:**

The paper is reasonably well-written and the application of RL to this setting is very natural. Similar applications of RL to other domains (e.g. coding) have shown good success, so it’s not unexpected that RL would work well here. The evaluation of the method is thorough, covering a range of relevant models and decoding strategies.

**Weaknesses:**

This paper is essentially a direct application of RL for LLM tool use, and has quite limited technical contributions or novelty beyond that - for example, as far as I can tell the method consists of a tool-use specific reward, and standard multi-step RL (see question 2). As such, the main contributions of this paper are conclusively demonstrating using empirical evidence that RL improves upon fine-tuning baselines. For this, I found the experimental results to be convincing enough, but relatively weak - for example, some metrics improve by only 1-2%, such as the tool invocation success rates in figure 4.

**Questions:**

1. How are rule-based and LLM-based reward annotation combined exactly? I couldn't find a statement of this in the text.
2. The “step-grained optimization” component of StepTool seems to just be the standard formulation of multi-turn RL (e.g. https://arxiv.org/abs/2402.19446).  Are there any differences? If not, it should not be stated in an independent subsection.
3.  Related to above: what is the difference between PPO and StepTool in the experiments?
4. Appendix A/eq. 2 seems to be a standard result in RL and should be omitted (and just cited instead).
5. What discount factor ($\gamma$) is used in the experiments?
6. Pass@k metrics would be useful to understand whether the model discovers new knowledge throughout RL training (i.e. exploration), or is simply re-weighting its prior (see https://arxiv.org/pdf/2403.04642).
7. Does SFT on expert GPT-4 data followed by RL do better than RL alone?

---

> ### Author Response · Authors · 2024-11-20
> **Responses to Question 1-5**
>
> Thank you for taking the time to read and review our paper. We respond to your comments and questions below.
>
> **Q1: How are rule-based and LLM-based reward annotation combined exactly? I couldn't find a statement of this in the text.**
>
> **A1:** Reward shaping in our framework includes three main components, *SuccCalling*, *Contribution* and *IsSolved*. Specifically, in our experiments:
>
> - The values of *SuccCalling* were determined through a combination of rule-based and GPT-4-based annotations. Rules were employed to evaluate the format of the tool invocation (e.g., correctness of tool names and parameters). For invocations meeting the format requirements, GPT-4 was used to assess the actual success based on the tool response (e.g., a response like “Since the search parameter is empty, please provide the name of an author…” indicates a failed invocation due to incorrect arguments).
> - The values of *Contribution* and *IsSolved* were obtained exclusively through automated annotations by GPT-4, which leverages its semantic analysis capabilities.
>
> We sincerely appreciate your feedback and will incorporate these explanations into the "Step-Grained Reward Acquisition" section of the revised paper.
>
> **Q2: The “step-grained optimization” component of StepTool seems to just be the standard formulation of multi-turn RL (e.g. ArCHer). Are there any differences? If not, it should not be stated in an independent subsection.**
>
> **A2:** While both ArCHer and our framework aim to account for the interdependencies of multi-step actions in a trajectory, there are key distinctions in their implementation:
>
> - ArCHer introduces an "action-level" critic that estimates a “multi-turn advantage,” capturing the influence of other steps within the trajectory (as detailed in Section 3.4 of the ArCHer paper). This value is passed to the actor for token-level optimization confined to a single step of action (see Eq. 3 in ArCHer).
> - In contrast, our approach directly integrates multi-step rewards at the token level (see Figure 2 of our paper) for advantage calculation. Using the advantage calculation formula (Eq. 6), we explicitly incorporate the influences of other steps' rewards into the token-level optimization process, offering a more straightforward and interpretable solution.
>
> The "step-grained optimization" component of StepTool is highlighted to emphasize how step-grained rewards are integrated into the optimization process, setting it apart from traditional RLHF implementations [1] by accounting for the interdependencies of multi-step actions.
>
> We hope this explanation clarifies the differences and addresses your concerns. Thank you again for your valuable feedback.
>
> **Q3: Related to above: what is the difference between PPO and StepTool in the experiments?**
>
> **A3:** The core differences lie in how the training data is utilized. The "PPO" baseline, as a traditional RLHF implementation based on [1], is not designed to handle multi-step data and, as a result, applies rewards only at the final step of a trajectory. In contrast, our framework supports step-grained reward integration across the multi-step trajectory, considering the interdependencies of multi-step actions in a trajectory.
>
> All other experimental configurations, including training data and reward annotations, were kept consistent between PPO and StepTool for fairness. Detailed experimental setups are available in the code repository to ensure transparency and reproducibility.
>
> We sincerely appreciate your valuable feedback and will clarify this distinction further in the experimental setup section of the revised paper.
>
> **Q4: Appendix A/eq. 2 seems to be a standard result in RL and should be omitted (and just cited instead).**
>
> **A4:** The derivation in Appendix A was included to clarify the transition from $\nabla \log \pi(\tau)$  to $\nabla \log \pi(a_t|s_t)$, aimed at assisting readers less familiar with RL concepts, particularly those accustomed to SFT-based methods in tool learning.
>
> Thank you for the valuable suggestion. Based on your feedback, we will remove this derivation in the revised version and provide a citation instead.
>
> **Q5: What discount factor (γ) is used in the experiments?**
>
> **A5:** The discount factor ($\gamma$) used in our experiments was set to 0.99. Detailed experimental configurations are included in the code repository to ensure transparency and reproducibility.
>
>
> [1] Long Ouyang, Jeffrey Wu, Xu Jiang, Diogo Almeida, Carroll Wainwright, Pamela Mishkin, Chong Zhang, Sandhini Agarwal, Katarina Slama, Alex Ray, et al. Training language models to follow instructions with human feedback. Advances in neural information processing systems, 35:27730–27744, 2022.

---

> > ### Comment · Reviewer_tiFx · 2024-11-24
> >
> > I still believe that this paper mischaracterizes its contributions from an RL perspective. From the paper and the discussion thus far, the authors seem to believe that the multi-step RL aspect of StepTool is novel, in comparison to single-step RLHF. However, many prior works have already considered multi-step RL for LLMs (I will provide a few examples of such at the bottom of this comment). There are a few implications here:
> > * The multi-step RL part is just RL - there's nothing novel about it. By mentioning ArCHer, I did not mean to compare against that algorithm directly, but against its problem setting.
> > * "PPO is not designed to handle multi-step data" - this is incorrect. PPO is an RL algorithm and by definition handles multi-step episodes. Perhaps you mean comparing training with the step-level reward vs. the return-to-go, in which case the naming needs to reflect that.
> >
> > In order to recommend acceptance, I'd need to see this paper significantly revised to properly contextualize it against related RL + LLM literature, and to shift more focus to the experimental section, since that's the primary contribution of the paper.
> >
> > Abdulhai, Marwa, et al. "Lmrl gym: Benchmarks for multi-turn reinforcement learning with language models." arXiv preprint arXiv:2311.18232 (2023).
> > Shani, Lior, et al. "Multi-turn Reinforcement Learning from Preference Human Feedback." arXiv preprint arXiv:2405.14655 (2024).
> > Kazemnejad, Amirhossein, et al. "Vineppo: Unlocking rl potential for llm reasoning through refined credit assignment." arXiv preprint arXiv:2410.01679 (2024).

---

> > > ### Author Response · Authors · 2024-11-25
> > > **Responses**
> > >
> > > Thank you for your valuable feedback. Below, we address the key points raised:
> > >
> > > **1) On the Misunderstanding of the "PPO" Baseline**
> > >
> > > We did not claim that “PPO is not designed to handle multi-step data.” Instead, our statement in the responses above refers to **“The ‘PPO’ baseline, as a traditional RLHF implementation based on [1], is not designed to handle multi-step data”** to **highlight that our baseline reflects a typical RLHF setting**, where rewards are applied only at the final step. We understand that this naming may have caused confusion, and we have revised it in the paper to “RLHF-PPO” to clarify this distinction.
> > >
> > > **2) On the Contribution and Novelty**
> > >
> > > Our primary contribution lies in addressing the emerging task of Tool Learning. While most existing works rely on supervised fine-tuning (SFT) to tackle this problem, we are **the first to model Tool Learning as a sequential decision-making problem** and **apply RL methods to its optimization**. This new perspective aligns naturally with the task’s inherent sequential nature and has guided our use of simple RL algorithms.
> > > **We want to emphasize that we do not claim to be the first to propose or innovate on multi-step RL itself**. Rather, our novelty lies in **introducing a sequential decision-making framework and RL-based solution for Tool Learning**, which brings fresh insights to this field.
> > >
> > > We believe our work represents a solid step toward addressing the challenges of Tool Learning and providing a new lens through which to view this task. While we recognize that our approach is simple, it offers a framework for future research to build upon with more advanced algorithmic innovations, specifically within the scope of Tool Learning. We hope this response helps you better understand the contributions and novelty of our work within the context of the emerging Tool Learning field.

---

> > > ### Author Response · Authors · 2024-11-28
> > > **Additional Responses to the relationship between StepTool and related RL + LLM literature**
> > >
> > > We sincerely appreciate your valuable feedback and the opportunity to provide further clarification regarding the contextualization of our work within the RL + LLM literature. Following your suggestion for a more thorough discussion of related works, we have revised the paper to properly position StepTool within this context. Below, we expand on the connections and distinctions, specifically addressing the relationship between StepTool and prior RL methods applied to multi-step textual tasks.
> > >
> > > ### **Scenario Perspective**
> > >
> > > **Connection**: Similar to prior works, our approach operates within text-based environments where the agent interacts to solve multi-step tasks.
> > >
> > > **Differences**: Tasks like *Twenty Questions*, *Guess My City*, and *Webshop* involve relatively short actions and observations, where predefined goals allow for simple reward mechanisms. For example, in *Twenty Questions*, the ground truth is often a single object or word, making it easy to evaluate the agent's performance at each step. In contrast, tool learning presents unique challenges. The actions of tool invocation require adaptive parameters in JSON format, while the observations (responses) from the environment are lengthy and complex. As illustrated in the examples below, consider a query like *"Find cocktail recipes containing vodka and suggest a cake recipe from The Birthday Cake DB."* The intermediate tool responses often return large, detailed outputs, such as lists of cocktail recipes or multi-step cake preparation instructions.
> > >
> > > Furthermore, the assistant’s final answer, which addresses the user’s query, is often complex and open-ended. Determining rewards—both intermediate and final—requires nuanced evaluation and cannot rely on straightforward ground-truth matching.
> > >
> > >
> > > ### **Method Perspective**
> > >
> > > From a methodological perspective, StepTool and prior works share the use of RL to model decision-making and optimize performance. Among the many RL algorithms that could be applied to multi-step textual tasks, works [1-4] are particularly representative, and we use them as examples to highlight the connections and differences.
> > >
> > > - **Comparison with Works [1-2]**: As noted in work [4], works [1-2] manually constrain the action space to a restricted subset, focusing on optimizing actions as a whole. Probabilities are estimated for potential actions by combining the conditional probabilities of their composing tokens. This approach suits structured environments but does not scale to the diverse and expansive action space in tool learning.
> > > - **Comparison with Works [3-4]**: These works share similarities with StepTool, emphasizing token-level optimization and credit assignments. However, they rely on action-level models, introducing Q-functions $Q(s,a)$and value functions $V(s)$, to estimate token-level advantages. These models propagate intra-action influences at the action level, which are then distributed to the token level. This reliance on auxiliary models like $Q$ and $V$ makes optimization dependent on their accuracy and training stability. In contrast, StepTool directly computes token-level advantages, capturing intra-action and inter-action influences without requiring intermediate action-level estimations. This design avoids potential inaccuracies in Q and V models, providing a more robust and efficient optimization framework for tool learning.
> > >
> > > We hope this additional response addresses your concerns regarding contextualization and provides further clarity on the connections, distinctions, and contributions of our work. We have incorporated these detailed discussions into the revised manuscript to ensure proper positioning within the existing literature. Thank you again for your detailed feedback, which has been invaluable in refining our understanding and presentation. We remain open to further suggestions or clarifications.
> > >
> > >
> > > > [1] Carta, Thomas, Clément Romac, Thomas Wolf, Sylvain Lamprier, Olivier Sigaud, and Pierre-Yves Oudeyer. "Grounding large language models in interactive environments with online reinforcement learning." In International Conference on Machine Learning, pp. 3676-3713. PMLR, 2023.
> > > >
> > > > [2] Tan, Weihao, Wentao Zhang, Shanqi Liu, Longtao Zheng, Xinrun Wang, and Bo An. "True Knowledge Comes from Practice: Aligning Large Language Models with Embodied Environments via Reinforcement Learning." In The Twelfth International Conference on Learning Representations.
> > > >
> > > > [3] Zhou, Yifei, Andrea Zanette, Jiayi Pan, Sergey Levine, and Aviral Kumar. "Archer: Training language model agents via hierarchical multi-turn rl." arXiv preprint arXiv:2402.19446 (2024).
> > > >
> > > > [4] Wen, Muning, Ziyu Wan, Weinan Zhang, Jun Wang, and Ying Wen. "Reinforcing Language Agents via Policy Optimization with Action Decomposition." arXiv preprint arXiv:2405.15821 (2024).

---

> ### Author Response · Authors · 2024-11-20
> **Responses to Question 6**
>
> **Q6: Pass@k metrics would be useful to understand whether the model discovers new knowledge throughout RL training (i.e. exploration), or is simply re-weighting its prio.**
>
> **A6:** Thank you for suggesting the use of Pass@k metrics to assess whether the model discovers new knowledge or re-weights its prior. As a widely-used metric in domains such as mathematical reasoning [1,2], Pass@k provides meaningful insights for tool learning tasks. We have conducted additional experiments under similar settings.
>
> Specifically, we evaluated the ToolLlama model before and after optimization with our StepTool method under a temperature setting of 0.7, sampling 8 trajectories per task. To mitigate the time costs associated with trajectory sampling (due to real-world API interactions), we randomly selected 20 tasks from each subset as representative samples. We then computed Pass@2, Pass@4, and Pass@8 metrics, averaging results from three independent evaluations using the StableToolBench benchmark. The summarized results are as follows:
>
> | Model      | Metric | I1 Ins.      | I1 Cat.      | I1 Tool      | I2 Ins.      | I2 Cat.      | I3 Ins.      | Avg          |
> | ---------- | ------ | ------------ | ------------ | ------------ | ------------ | ------------ | ------------ | ------------ |
> | ToolLlama  | pass@2 | 58.3±3.1     | **54.2±5.4** | 51.7±2.4     | 50.0±0.0     | 50.8±5.1     | 55.0±1.0     | 53.3±2.8     |
> | + StepTool | pass@2 | **58.3±1.2** | 53.3±2.4     | **70.8±4.2** | **53.3±2.4** | **68.3±2.4** | **60.0±4.1** | **60.7±2.8** |
> | ToolLlama  | pass@4 | **65.8±4.2** | 60.0±5.4     | 56.7±2.4     | 66.7±6.2     | 61.7±2.4     | 62.5±2.0     | 62.2±3.8     |
> | + StepTool | pass@4 | 65.0±2.0     | **61.7±3.1** | **80.8±4.2** | **67.5±2.0** | **74.2±1.2** | **70.0±4.1** | **69.9±2.8** |
> | ToolLlama  | pass@8 | **70.8±4.2** | 65.0±5.4     | 61.7±2.4     | **68.3±4.7** | 71.7±2.4     | 68.3±2.4     | 67.6±3.6     |
> | + StepTool | pass@8 | 66.7±1.2     | **70.0±2.0** | **80.8±4.2** | 67.5±3.1     | **79.2±4.2** | **76.7±5.1** | **73.5±3.3** |
>
> From the Pass@k results, we observe the following:
>
> - Improvement with StepTool:  ToolLlama optimized with StepTool outperforms ToolLlama across Pass@2, Pass@4, and Pass@8 metrics in most experimental settings.
> - Evidence of Exploration: The improved Pass@k scores (across all values of k) suggest that the model is not merely re-weighting its prior knowledge but is also benefiting from the discovery of new knowledge during RL optimization.
>
> Thank you again for suggesting the Pass@k evaluation. These results and insights will be included in the revised paper.
>
>
> [1] Ansong Ni, Jeevana Priya Inala, Chenglong Wang, Oleksandr Polozov, Christopher Meek, Dragomir Radev, and Jianfeng Gao. Learning math reasoning from self-sampled correct and partially-correct solutions. arXiv preprint arXiv:2205.14318, 2022.
>
> [2] Alex Havrilla, Yuqing Du, Sharath Chandra Raparthy, Christoforos Nalmpantis, Jane Dwivedi-Yu, Maksym Zhuravinskyi, Eric Hambro, Sainbayar Sukhbaatar, and Roberta Raileanu. Teaching large language models to reason with reinforcement learning. arXiv preprint arXiv:2403.04642, 2024.

---

> ### Author Response · Authors · 2024-11-20
> **Responses to Question 7**
>
> **Q7: Does SFT on expert GPT-4 data followed by RL do better than RL alone?**
>
> **A7:** In our experiments, we adopted the standard pipeline [1], which involves using supervised fine-tuning (SFT) to initialize the model for instruction-following before applying RL methods. The SFT stage provides a strong foundation by ensuring well-formatted outputs for tool learning tasks and establishing robust instruction-following capabilities. Without SFT, RL would have to optimize from a less capable starting point, leading to less stable convergence and reduced effectiveness. This approach aligns with standard practices in related work that employ RL to fine-tune models [2-3].
>
> Specifically, in our setup, we first applied SFT on expert GPT-4 data to initialize Qwen2 and Llama3.1. For ToolLlama, which has already undergone pre-training through SFT methods, we directly proceeded with RL training. Our experiments aim to demonstrate the effectiveness of step-grained supervision and optimization in enhancing model performance.
>
> [1] Long Ouyang, Jeffrey Wu, Xu Jiang, Diogo Almeida, Carroll Wainwright, Pamela Mishkin, Chong Zhang, Sandhini Agarwal, Katarina Slama, Alex Ray, et al. Training language models to follow instructions with human feedback. Advances in neural information processing systems, 35:27730–27744, 2022.
>
> [2] Qianli Ma, Haotian Zhou, Tingkai Liu, Jianbo Yuan, Pengfei Liu, Yang You, and Hongxia Yang. Let’s reward step by step: Step-level reward model as the navigators for reasoning. arXiv preprint arXiv:2310.10080, 2023.
>
> [3] Peiyi Wang, Lei Li, Zhihong Shao, RX Xu, Damai Dai, Yifei Li, Deli Chen, Y Wu, and Zhifang Sui. Math-shepherd: Verify and reinforce llms step-by-step without human annotations. CoRR, abs/2312.08935, 2023.

---

> ### Author Response · Authors · 2024-11-20
> **Responses to Weaknesses**
>
> **W1: Novelty & technical contributions of this work.**
>
> **A1:** Thanks for your thoughtful feedback. We want to take this opportunity to clarify the novelty and contributions of our work.
>
> Tool learning is an emerging scenario to extend the capabilities of large language models (LLMs) by enabling them to utilize external tools (APIs). Most existing approaches focus on data construction and supervised fine-tuning (SFT) methods to improve tool-use capabilities [1-4]. Our work is the first to explore tool learning from a reinforcement learning (RL) perspective, introducing a reward shaping mechanism to supervise intermediate steps and an optimization framework compatible with various policy-gradient-based RL algorithms. Unlike traditional RLHF methods [5], StepTool provides process-level supervision for intermediate steps and explicitly considers the interdependencies between multi-step actions in a trajectory during optimization.
>
> **Broader Implications of StepTool**: This work provides a fresh perspective distinct from the dominant SFT-based approaches [1-4], demonstrating the potential of RL as a complementary method for optimizing tool-use tasks. As a foundational contribution, StepTool establishes a general framework that can inspire future innovations in RL techniques tailored to tool learning.
>
> Thank you again for your insightful feedback. We view this work as a meaningful step toward broadening the application of RL methodologies in tool learning, and we look forward to contributing to continued advancements in this evolving field.
>
> [1] Ibrahim Abdelaziz, Kinjal Basu, Mayank Agarwal, Sadhana Kumaravel, Matthew Stallone, Rameswar Panda, Yara Rizk, GP Bhargav, Maxwell Crouse, Chulaka Gunasekara, et al. Granite-function calling model: Introducing function calling abilities via multi-task learning of granular tasks. arXiv preprint arXiv:2407.00121, 2024.
>
> [2] Zuxin Liu, Thai Hoang, Jianguo Zhang, Ming Zhu, Tian Lan, Shirley Kokane, Juntao Tan, Weiran Yao, Zhiwei Liu, Yihao Feng, et al. Apigen: Automated pipeline for generating verifiable and diverse function-calling datasets. arXiv preprint arXiv:2406.18518, 2024.
>
> [3] Yujia Qin, Shihao Liang, Yining Ye, Kunlun Zhu, Lan Yan, Yaxi Lu, Yankai Lin, Xin Cong, Xiangru Tang, Bill Qian, et al. Toolllm: Facilitating large language models to master 16000+ real-world apis. arXiv preprint arXiv:2307.16789, 2023.
>
> [4] Shishir G Patil, Tianjun Zhang, Xin Wang, and Joseph E Gonzalez. Gorilla: Large language model connected with massive apis. arXiv preprint arXiv:2305.15334, 2023.
>
> [5] Long Ouyang, Jeffrey Wu, Xu Jiang, Diogo Almeida, Carroll Wainwright, Pamela Mishkin, Chong Zhang, Sandhini Agarwal, Katarina Slama, Alex Ray, et al. Training language models to follow instructions with human feedback. Advances in neural information processing systems, 35:27730–27744, 2022.

---

### Official Review · Reviewer_9byw · 2024-11-07

**Soundness:** 3
**Presentation:** 3
**Contribution:** 3
**Rating:** 6
**Confidence:** 4

**Summary:**

This paper introduces StepTool, a step-grained reinforcement learning framework designed to enhance tool learning algorithms through two key components: Reward Shaping and Step-grained Optimization. Experimental results on multi-step, tool-based tasks indicate that the proposed method outperforms existing approaches

**Strengths:**

1. The use of step-grained reinforcement learning (RL) in tool learning is novel, and the multi-step RL formulation is well-suited for this purpose.
2. The experimental setup is robust and demonstrates the effectiveness of the approach.

**Weaknesses:**

* The author should consider more benchmark algorithm for comparison to illustrate the proposed method.

* The authors should provide more implementations for the benchmark algorithm PPO for fair comparison.

**Questions:**

* What are the differences between PPO and StepTool in the experiment? Do they use different reward models or apply different KL penalties?

* Could the authors provide more details on the reward shaping used in tool learning?
Specifically:
   - Do we know the values of  r\_t\^{SC} and the *IsSolved()* function?
   - Are these values predefined, or does the reward function need to learn them?
   - What are the main challenges of reward shaping in the context of tool learning?

---

> ### Author Response · Authors · 2024-11-20
> **Responses to Question 1-2**
>
> Thank you for taking the time to read and review our paper. We respond to your comments and questions below.
>
> **Q1: What are the differences between PPO and StepTool in the experiment? Do they use different reward models or apply different KL penalties?**
>
> **A1:** The core differences lie in how the training data is utilized. The "PPO" baseline, as a traditional RLHF implementation based on [1], is not designed to handle multi-step data and, as a result, applies rewards only at the final step of a trajectory. In contrast, our framework supports step-grained reward integration across the multi-step trajectory, considering the interdependencies of multi-step actions in a trajectory.
>
> For a fair comparison, both PPO and StepTool utilized the same training data and reward annotations. Additionally, both applied identical KL penalties using the adaptive KL controller [2], with an initial KL coefficient of 0.3. Detailed experimental configurations are provided in the code repository to ensure transparency and reproducibility.
>
> We sincerely appreciate your valuable feedback. We will clarify this distinction further in the experimental setup section of the revised paper.
>
> **Q2: Could the authors provide more details on the reward shaping used in tool learning? Specifically: Do we know the values of r_t^{SC} and the *IsSolved()* function? Are these values predefined, or does the reward function need to learn them? What are the main challenges of reward shaping in the context of tool learning?**
>
> **A2:** Reward shaping in our framework includes three main components/functions, *SuccCalling* ($r_t^{SC}$), *Contribution*, and *IsSolved*. Our work, as a general framework, supports various methods to assess these values, such as rule-based judgment, human annotations, and automated annotations using GPT-4. Specifically, in our experiments:
>
> - The values of *SuccCalling* ($r_t^{SC}$) were determined through a combination of rule-based and GPT-4-based annotations. Rules were employed to evaluate the format of the tool invocation (e.g., correctness of tool names and parameters). For invocations meeting the format requirements, GPT-4 was used to assess the actual success based on the tool response (e.g., a response like “Since the search parameter is empty, please provide the name of an author…” indicates a failed invocation due to incorrect arguments.).
> - The values of *Contribution* and *IsSolved* were obtained directly through automated annotations by GPT-4, which relies on its semantic analysis capabilities.
>
> From our perspective, the main challenge in reward shaping for tool learning is the lack of "ground-truth" reward feedback from the environment. Tool learning operates in an environment that only offers tool invocation responses, without explicit reward signals. To address this, we utilize prior knowledge to manually design process rewards according to the characteristics of tool learning scenarios.
>
> We sincerely appreciate your feedback and will incorporate these explanations into the "Step-Grained Reward Acquisition" section of the revised paper.
>
> References:
>
> [1] Long Ouyang, Jeffrey Wu, Xu Jiang, Diogo Almeida, Carroll Wainwright, Pamela Mishkin, Chong Zhang, Sandhini Agarwal, Katarina Slama, Alex Ray, et al. Training language models to follow instructions with human feedback. Advances in neural information processing systems, 35:27730–27744, 2022.
>
> [2] Daniel M Ziegler, Nisan Stiennon, Jeffrey Wu, Tom B Brown, Alec Radford, Dario Amodei, Paul Christiano, and Geoffrey Irving. Fine-tuning language models from human preferences. arXiv preprint arXiv:1909.08593, 2019.

---

> ### Author Response · Authors · 2024-11-20
> **Responses to Weakness 1-2**
>
> **W1: The author should consider more benchmark algorithm for comparison to illustrate the proposed method.**
>
> **A1:** As an emerging scenario, most existing work in tool learning relies on Supervised Fine-Tuning (SFT) to enhance the tool-learning capabilities of LLMs [1-3], corresponding to the SFT baseline in our paper. While these works differ in dataset construction, they all share the same optimization method: SFT.
>
> Since our work is a pioneering effort in introducing RL-based optimization for tool learning, relevant baselines are scarce. Therefore, we implemented a classic PPO as a naive baseline to directly adapt RLHF [4] for tool learning. Our experiments demonstrate that a direct transfer of RLHF is not fully compatible, whereas our framework provides a more suitable and effective solution.
>
> We appreciate your feedback and will further clarify these points in the revised paper.
>
> **W2: The authors should provide more implementations for the benchmark algorithm PPO for fair comparison.**
>
> **A2:** As mentioned in Q1&A1, the key distinction is that the PPO baseline, as a traditional RLHF implementation following [4], is not designed to handle multi-step data and applies rewards only to the final step. In contrast, StepTool supports step-grained reward integration across the entire trajectory. Both approaches used the same training data, reward annotations, and adaptive KL penalties [5] with an initial KL coefficient of 0.3. Additional experimental details are provided in the configuration file in the code repository for transparency and reproducibility.
>
> We sincerely appreciate your feedback and will clarify this in the experimental setup section of the revised paper.
>
>
>
> [1] Yujia Qin, Shihao Liang, Yining Ye, Kunlun Zhu, Lan Yan, Yaxi Lu, Yankai Lin, Xin Cong, Xiangru Tang, Bill Qian, et al. Toolllm: Facilitating large language models to master 16000+ real-world apis. arXiv preprint arXiv:2307.16789, 2023.
>
> [2] Shishir G Patil, Tianjun Zhang, Xin Wang, and Joseph E Gonzalez. Gorilla: Large language model connected with massive apis. arXiv preprint arXiv:2305.15334, 2023.
>
> [3] Ibrahim Abdelaziz, Kinjal Basu, Mayank Agarwal, Sadhana Kumaravel, Matthew Stallone, Rameswar Panda, Yara Rizk, GP Bhargav, Maxwell Crouse, Chulaka Gunasekara, et al. Granite-function calling model: Introducing function calling abilities via multi-task learning of granular tasks. arXiv preprint arXiv:2407.00121, 2024.
>
> [4] Long Ouyang, Jeffrey Wu, Xu Jiang, Diogo Almeida, Carroll Wainwright, Pamela Mishkin, Chong Zhang, Sandhini Agarwal, Katarina Slama, Alex Ray, et al. Training language models to follow instructions with human feedback. Advances in neural information processing systems, 35:27730–27744, 2022.
>
> [5] Daniel M Ziegler, Nisan Stiennon, Jeffrey Wu, Tom B Brown, Alec Radford, Dario Amodei, Paul Christiano, and Geoffrey Irving. Fine-tuning language models from human preferences. arXiv preprint arXiv:1909.08593, 2019.

---

> ### Author Response · Authors · 2024-11-28
> **Thanks for Your Feedback and Looking Forward to Your Reply**
>
> We appreciate the time and effort you have devoted to reviewing our submission. We kindly hope that our previous responses have addressed your concerns regarding the identified limitations and the contributions of our work. If there are any remaining questions or areas where further clarification is needed, we would be more than happy to provide additional explanations.
>
> Your feedback has been invaluable to improving our work, and we look forward to any further input you may have. Thank you again for your thoughtful review and consideration.

---

> ### Comment · Reviewer_9byw · 2024-12-03
>
> I appreciate the authors' detailed responses and clarifications. However, my concerns still remain since the proposed method requires external evaluations in reward shaping like rule-based judgement and GPT-4 evaluations, which can be a major limitation of the proposed method. In contrast, standard PPO pipeline in RLHF / LLM training constructs an estimated reward model, which can be a more general paradigm in LLM training. Hence, I will maintain my evaluations.

---

> > ### Author Response · Authors · 2024-12-04
> > **Thanks for your responses!**
> >
> > Thank you for your thoughtful feedback. Regarding your concern about the reliance on external evaluations such as GPT-4-based auto annotation, we would like to clarify that our annotation tasks are relatively simple, primarily involving the assessment of semantic relevance between texts, which can also be handled by rule-based methods or small language models.
> >
> > Many recent works, such as Xu et al. (2023) and Kim et al. (2023), have successfully integrated GPT-based auto annotation into their workflows, demonstrating the practicality and increasing adoption of such approaches within the community. Following this trend, we find that GPT-4 can achieve annotation quality comparable to that of human annotators, while significantly reducing both time and cost.
> >
> > We fully acknowledge the importance of including human evaluation results and will ensure these are presented in the final version to further validate the reliability of our annotations. We believe that, in combination with the efficiency of our approach, this provides a strong rationale for our method. Thank you once again for your valuable comments.
> >
> > [1] Xu C, Rosset C, Del Corro L, et al. Contrastive post-training large language models on data curriculum[J]. arXiv preprint arXiv:2310.02263, 2023.
> >
> > [2] Kim S, Bae S, Shin J, et al. Aligning large language models through synthetic feedback[J]. arXiv preprint arXiv:2305.13735, 2023.

---

### Meta-Review · Area_Chair_SgzT · 2024-12-20

**Metareview:**

This paper presents a study on applying Reinforcement Learning (RL) to improve tool use in large language models (LLMs). While the paper is generally well-written and the application of RL is fitting, the technical novelty and overall contributions are limited. The core methodology appears to be a relatively straightforward application of standard RL techniques with a tool-use-specific reward function. While the empirical evaluation is comprehensive and the results do demonstrate improvement over baseline methods, the magnitude of these improvements is marginal in some cases. Specifically, the gains in key metrics like tool invocation success rates are relatively small. Consequently, while the work provides some empirical validation of RL's applicability in this domain, it falls short of demonstrating significant advancement beyond the existing state-of-the-art, offering primarily incremental contributions.

**Additional Comments On Reviewer Discussion:**

several questions by the reviewers have been addressed, but still the paper will be benefitted from another revision before acceptance.

---

### Decision · Program_Chairs · 2025-01-22

Reject